systems biology/computational biology/
mathematical modelling

metabolic oscillations, *Bacillus subtilis*,
minimal model, biofilm, Hopf bifurcation,
glutamate metabolism

**Author for correspondence:**
Stefan Schuster
e-mail: stefan.schu@uni-jena.de

# Differential equation-based minimal model describing metabolic oscillations in *Bacillus subtilis* biofilms

Ravindra Garde[1,2], Bashar Ibrahim[1,4,5], Ákos T. Kovács[3] and Stefan Schuster[1]

[1]Department of Bioinformatics, Matthias Schleiden Institute, Friedrich Schiller University Jena, Ernst-Abbe-Platz 2, 07743 Jena, Germany
[2]Max Planck Institute for Chemical Ecology, Hans-Knöll-Strasse 8, 07745 Jena, Germany
[3]Bacterial Interactions and Evolution Group, DTU Bioengineering, Technical University of Denmark, Søltofts Plads Building 221, 2800 Kgs. Lyngby, Denmark
[4]Centre for Applied Mathematics and Bioinformatics, and [5]Department of Mathematics and Natural Sciences, Gulf University for Science and Technology, Hawally 32093, Kuwait

BI, 0000-0001-7773-0122; ÁTK, 0000-0002-4465-1636;
SS, 0000-0003-2828-9355

Biofilms offer an excellent example of ecological interaction among bacteria. Temporal and spatial oscillations in biofilms are an emerging topic. In this paper, we describe the metabolic oscillations in *Bacillus subtilis* biofilms by applying the smallest theoretical chemical reaction system showing Hopf bifurcation proposed by Wilhelm and Heinrich in 1995. The system involves three differential equations and a single bilinear term. We specifically select parameters that are suitable for the biological scenario of biofilm oscillations. We perform computer simulations and a detailed analysis of the system including bifurcation analysis and quasi-steady-state approximation. We also discuss the feedback structure of the system and the correspondence of the simulations to biological observations. Our theoretical work suggests potential scenarios about the oscillatory behaviour of biofilms and also serves as an application of a previously described chemical oscillator to a biological system.

# 1. Introduction

Development of a complex biofilm provides several benefits to bacteria, including efficient nutrient distribution, defence from chemical attacks or, in the case of a floating pellicle on the surface of liquids, better gaseous exchange [1]. Biofilms are thus complex communities of bacteria and as such, many types of

social dynamics come into play [2,3]. One of these is the division of labour [4,5]. The core of the biofilm growing on a solid surface shows a different metabolic state than the periphery. The periphery can freely access the nutrients from the surrounding environment. The interior, however, faces hindrance in obtaining a stable inflow of nutrients because the peripheral cells use up the nutrients that diffuse towards the interior. An experimental set-up to simulate that situation is provided by a microfluidics chamber [4].

An example of such a nutrient gradient is the production and diffusion of ammonia in the biofilm. Every cell in the biofilm has the ability to produce ammonia [4,6]. However, this small chemical compound is highly diffusive and therefore escapes into the environment as soon as it is produced by the cells in the periphery, thus leading to waste of nitrogen. In the interior, the ammonia produced by the cells diffuses out into the periphery. Thus, the interior cells monopolize ammonia production for the entire biofilm. Ammonia being an essential component of glutamine metabolism could be used to control the growth rate of the periphery by limiting its supply. The interplay between the inner and outer cells is required for glutamine synthesis and therefore the growth of the biofilm [4,6,7].

To understand biofilms more closely and make predictions based on empirical data, several models have been developed [4,7–12]. Liu *et al.* [4] observed oscillations in the biofilm, which they explained by different metabolic roles performed by the different compartments in the biofilm. They also established a model based on six differential equations. They defined two regions: the interior and periphery. Each of the regions has a variable representing glutamate and another representing the concentration of housekeeping proteins like ribosomal proteins. Ammonia and the active form of the enzyme glutamate dehydrogenase are also variables of the model.

Since many biological oscillators have been described by less than six variables [13,14], a simpler model could be established for biofilm oscillations as well. Our ultimate aim was to develop a minimal model to describe the metabolic oscillations happening in a biofilm. Minimal models are the simplest way to describe a certain phenomenon with the least number of parameters [15] and this is in agreement with Occam's razor. For example, minimal models were established for glycolytic oscillations by Higgins [16] and Sel'kov [17] and for calcium oscillations by Somogyi & Stucki [18].

Here, we employ the smallest chemical reaction system showing a Hopf bifurcation [19], which was further analysed [20,21] and used to describe p53 oscillations [22]. At a Hopf bifurcation, damped oscillations turn into undamped oscillations [15,23]. In particular, Wilhelm & Heinrich [19,20] performed a thorough stability analysis of the model. We test to what extent the terms in this model match the processes in a biofilm system. In this analysis, we focus on the Hopf bifurcation, discuss the feedback structure and point out the correspondence of the simulations to biological observations.

In our model, we use three variables only: ammonia, and interior and peripheral glutamate. Besides the quest for minimality, a reason for not considering the concentrations of housekeeping proteins as variables is that they change on a longer time-scale than metabolites. A similarity to the larger Liu model [4] is that, among the various amino acids, we focus on the metabolism of glutamate since glutamate and ammonia are both involved in the production of various amino acids through *trans*-amination, which is then equated to growth.

To study the effect and possible benefit of oscillations, it is of interest to compute the average values of variables, as was done for several oscillators [24–29]. For linear differential equation systems showing oscillations (such as the system describing the harmonic pendulum), the average values equal the values at the marginally stable steady state. For nonlinear differential equation systems, the average values often differ from the values at the unstable steady state surrounded by the oscillations. However, there are some types of nonlinear systems for which equality holds, for example, Lotka–Volterra systems of any dimension [25]. The equality property has also been proved for some models of calcium oscillations [27,30] and the Higgins–Selkov oscillator [27]. Here, we probe the model employed for describing biofilm oscillations for the above-mentioned property.

# 2. Material and methods

## 2.1. The model

Based on the scenario described by Liu *et al.* [4], the biofilm was separated into two compartments—the interior and the periphery. Here, we use a minimalist approach and try to model biofilm oscillations with the simplest model possible. Accordingly, we use the smallest chemical reaction system with Hopf

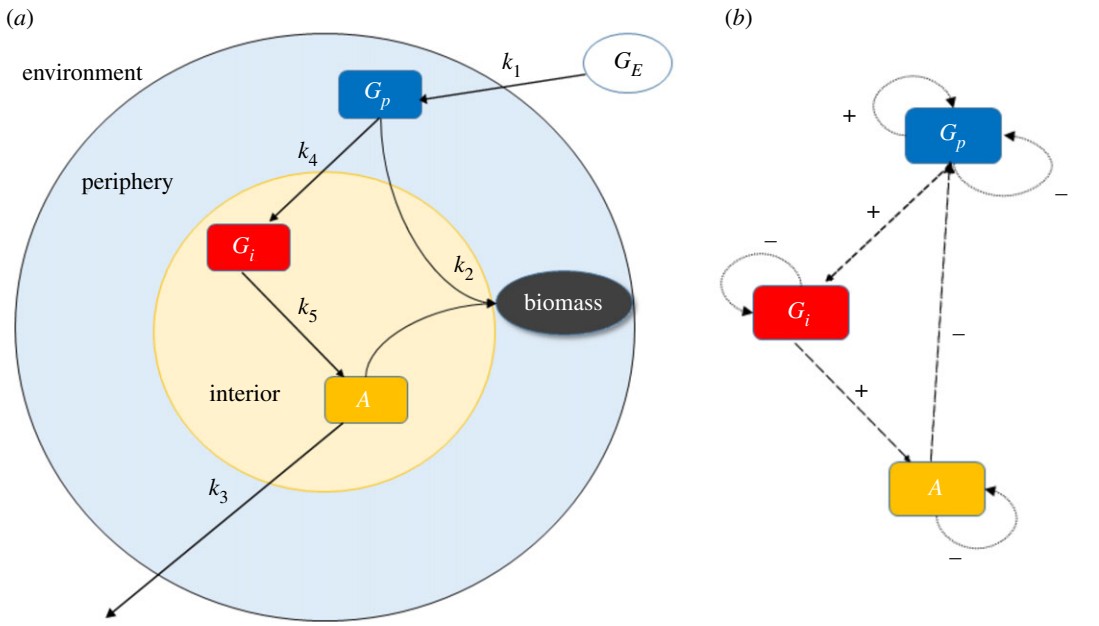

**Figure 1.** Schematic of the biofilm metabolic oscillation model. (*a*) The five reactions with rate constants $k_1$ through $k_5$ between the substances $G_i$, $G_p$, $A$ having variable concentrations and $G_E$ considered to be constant. The final result is the production of biomass. (*b*) Feedback structure of the model. $G_p$ is self-amplifying while all three variables are self-degrading. $G_p$ positively influences $G_i$, which positively influences $A$ which negatively influences $G_p$, thus, the overall feedback is negative.

bifurcation [19]. The term chemical system mathematically means that only up to bilinear terms are involved. It turns out that this model matches the biological set-up.

The model includes five reactions (figure 1) and three species with variable concentrations. The general variables $X$, $Y$ and $Z$ from the Wilhelm and Heinrich model can be assigned for the biofilm system to: peripheral glutamate ($G_p$), ammonia ($A$) and internal glutamate ($G_i$), respectively. Based on mass-action kinetics, the reactions have been translated as follows into a system of ordinary differential equations (ODEs):

$$\frac{\mathrm{d}G_p}{\mathrm{d}t} = k_1 G_E G_p - k_4 G_p - k_2 A G_p, \tag{2.1a}$$

$$\frac{\mathrm{d}A}{\mathrm{d}t} = -k_3 A + k_5 G_i \tag{2.1b}$$

and

$$\frac{\mathrm{d}G_i}{\mathrm{d}t} = k_4 G_p - k_5 G_i. \tag{2.1c}$$

Model assumptions and interpretation of terms in the model:

— $k_1 G_E G_p$: The uptake of glutamate from the environment ($G_E$) by the periphery of the biofilm. $G_E$ is supplied in a large excess, hence considered constant. The uptake of glutamate ($G_p$) is dependent on itself because glutamate represents the total amino acid and thus protein concentration in the biofilm periphery and can be assumed, in rough approximation, to be proportional to the concentration of various transport proteins embedded in the cell membranes. The greater the concentration of these proteins, the higher is the glutamate uptake rate. Without this self-amplification of glutamate, the system would not oscillate by construction of the minimal model.
— $k_4 G_p$: Diffusion of glutamate from the periphery of the biofilm into its interior. We do not consider self-amplification by $G_i$ in the main text. We analysed the effect of self-amplification of $G_i$ using the term $k_4 G_i G_p$ (electronic supplementary material, figure S4).
— $k_2 A G_p$: Consumption of glutamate and ammonia to produce biomass. As a simplification, we assumed that only the interior cells produce ammonia since that produced by the peripheral cells is rapidly lost to the environment.
— $k_5 G_i$: Consumption of glutamate to produce ammonia.
— $k_3 A$: Diffusion of ammonia into the surroundings. The loss of ammonia due to diffusion is much larger than that taken up by the periphery to produce biomass. Therefore, the term $k_2 A G_p$ does not appear in equation (2.1b).

**Table 1.** List of parameters used in the model. For explanations, see text.

| parameter | symbol | value with unit |
|---|---|---|
| rate constant of glutamate diffusion from environment to biofilm | $k_1$ | 0.3426 (mmol l$^{-1}$ h)$^{-1}$ |
| biomass formation coefficient | $k_2$ | 5.3 (mmol l$^{-1}$ h)$^{-1}$ |
| rate constant of ammonia diffusion [32] | $k_3$ | 4 h$^{-1}$ |
| rate constant of glutamate diffusion within biofilm [33,34] | $k_4$ | 2 h$^{-1}$ |
| ammonia production coefficient | $k_5$ | 2.3 h$^{-1}$ |
| glutamate concentration in the environment [4] | $G_E$ | 30 mmol l$^{-1}$ |
| conversion factor for biomass production | $b$ | 0.1 ((mmol l$^{-1}$)$^2$ h)$^{-1}$ |

## 2.2. Simulation

For computer simulations, we used the software COPASI v. 4.16 and 4.24 [31] and its LSODA deterministic solver. The simulations were double-checked using the Matlab ode15s (MathWorks) function. The figures of the simulations were produced using COPASI, and the three-dimensional (3D) phase plot was generated using the lines3D function of R plot3D library. The biomass plot was generated using the R function ggplot.

Parameter values are given in table 1. They are obtained by rescaling the parameter values from the Wilhelm & Heinrich paper [19] such that the oscillation period observed in the experimental work by Liu *et al.* [4] is matched. The glutamate concentration in the environment was adopted from the Liu *et al.* paper [4]. We have chosen the rate constant of diffusion of ammonia, $k_3$, to be twice as high as that of glutamate, $k_4$. This is because the diffusion coefficient for ammonia [32] is about $1.6 \times 10^{-5}$ cm$^2$ s$^{-1}$, while that for glutamate [33,34] is about $8 \times 10^{-6}$ cm$^2$ s$^{-1}$. $k_1$ and $k_2$ had to be increased in order to obtain undamped oscillations and to match the same period. Overall, the parameters allow a good comparison to the results by Wilhelm & Heinrich [19], while also being realistic from a physico-chemical point of view.

The predicted doubling time was calculated by averaging the relative increase in biomass at four consecutive time points of the maxima of ammonia concentration.

# 3. Results

## 3.1. Steady states

The steady states of the system can be calculated analytically. This gives a trivial steady state (TSS)

$$G_p = A = G_i = 0 \tag{3.1}$$

and a non-trivial steady state (NTSS)

$$G_{p,\text{ss}} = \left(\frac{k_1 G_E - k_4}{k_2 k_4}\right) k_3 \tag{3.2a}$$

$$A_{\text{ss}} = \frac{k_1 G_E - k_4}{k_2} \tag{3.2b}$$

and

$$G_{i,\text{ss}} = \left(\frac{k_1 G_E - k_4}{k_2 k_5}\right) k_3. \tag{3.2c}$$

It is worth noting that the concentrations at the latter state are linear functions of $G_E$. The TSS and NTSS are stable if $k_1 G_E - k_4$ is negative or positive, respectively [19]. At the threshold, a transcritical bifurcation [35] occurs; that is, the two steady states interchange their stability. At a further threshold, $k_1 G_E = k_3 + k_4 + k_5$, the stable NTSS turns unstable in a Hopf bifurcation [19].

## 3.2. Time course shows oscillations

We run the time course calculation of the system (2.1a–c) for 25 simulation hours with 1000 steps each of size 0.025 h (1.5 min). The period for oscillations is about 126 min (2 h 6 min), which is in agreement with

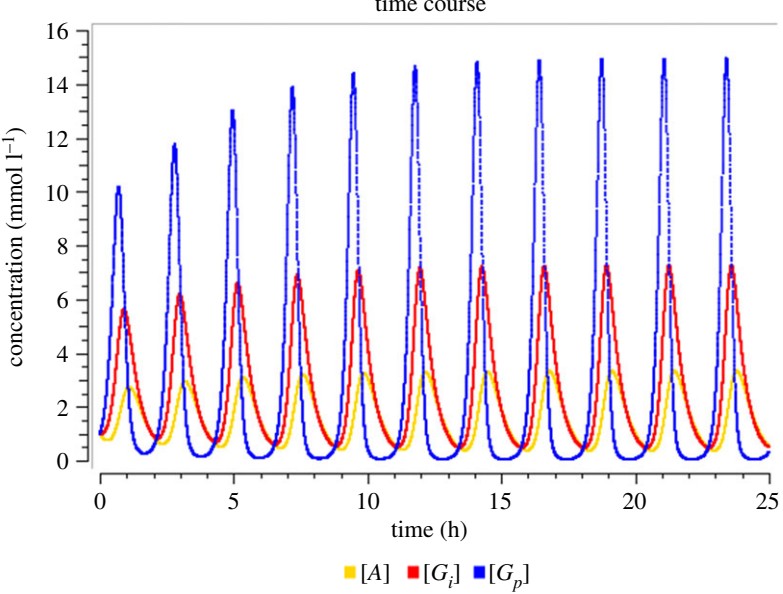

**Figure 2.** Time course of ammonia (yellow) and interior (red) and peripheral (blue) glutamate as computed by the minimal model. Parameter values: refer to table 1.

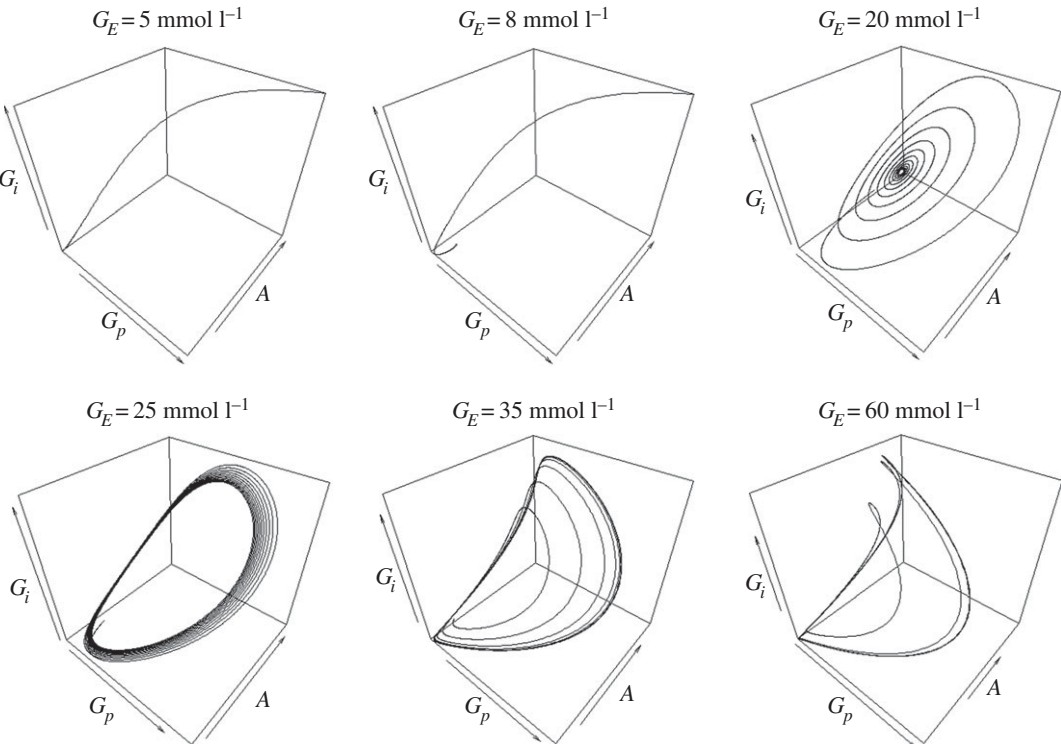

**Figure 3.** Three-dimensional phase portrait of all the variables at various values of $G_E$. The trajectory runs anti-clockwise in the perspective shown here. Top three trajectories from left to right depict approaches towards: the TSS, the NTSS in a non-oscillatory way and the NTSS by damped oscillations. The bottom three trajectories depict the convergence towards limit cycles beyond the Hopf bifurcation ($G_E = 24.41$). For parameter values except $G_E$, see table 1.

the experimental observations [4], because the parameters have been rescaled accordingly (see above). The amplitude of oscillations is observed to be 3.0 mmol $l^{-1}$ for ammonia, 7.1 mmol $l^{-1}$ for interior glutamate and 14.9 mmol $l^{-1}$ for peripheral glutamate (figure 2). It can be seen that the three variables oscillate with phase shifts, i.e. asynchronously.

In order to see the interdependence between the variables of our model, we plot the phase portrait of all three variables for various values of $G_E$ (figure 3). $G_E$ is an appropriate bifurcation parameter because

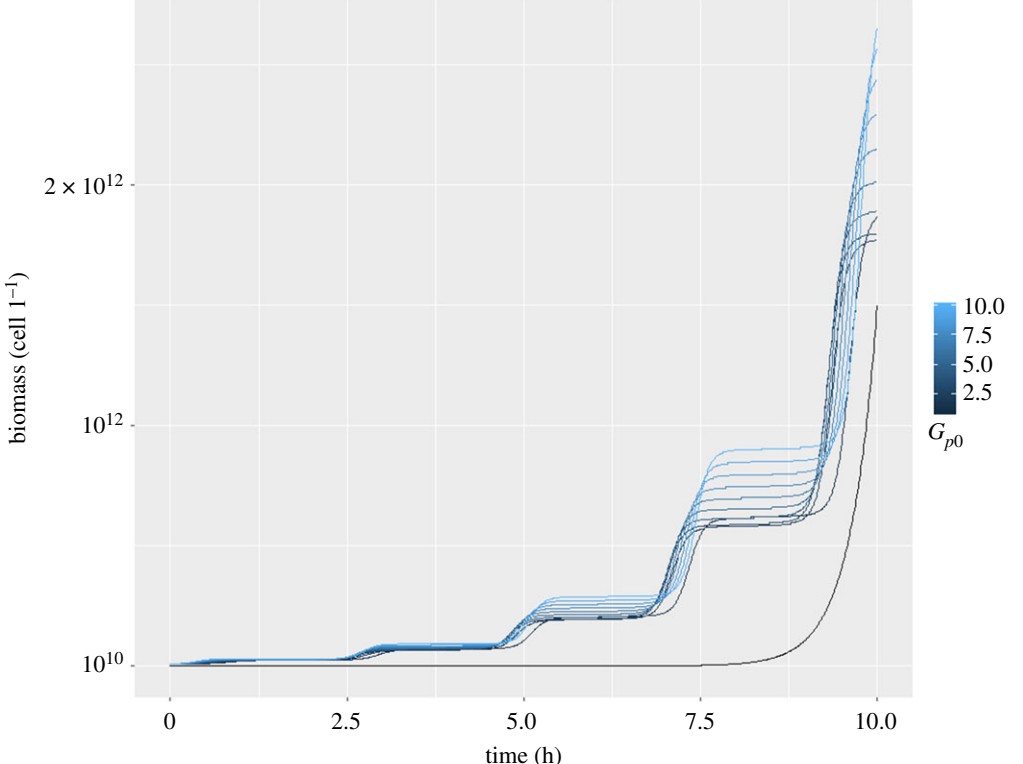

**Figure 4.** Plot of the time course of growth as calculated from equation (3.4) for various initial values of $G_p$ from 1 to 10 mmol $l^{-1}$ with a step size of 1 mmol $l^{-1}$ (all wavy curves). We took $10^{10}$ cells $l^{-1}$ as the initial value of $B$. On average, the curves have a doubling time of about 99 min. The black monotonic curve (initial value: 10 cells $l^{-1}$) indicates the growth calculated by the steady-state values.

the external glutamate concentration can be changed in experiment. Wilhelm & Heinrich [19] presented a similar figure for the vicinity of the Hopf bifurcation. In our figure 3, also the non-oscillatory relaxation towards the TSS and NTSS for appropriate parameter values is shown. $G_E$ values below 15 mmol $l^{-1}$ eliminate the oscillations (even damped ones).

As per the assumptions of our model, $k_2 A G_p$ is a proxy for the input for the synthesis of biomass from ammonia and glutamate and is thus related to the growth of the biofilm. Biomass production can be described by the following differential equation:

$$\frac{dB}{dt} = bAG_pB, \tag{3.3}$$

where $b$ is a conversion factor and was tuned to 0.1 $((mmol\ l^{-1})^2\ h)^{-1}$ so that the doubling time is in agreement with the experimental values [36]. The number of cells can be converted to biomass by taking the volume of a typical *B. subtils* cell of about $0.85 \pm 0.38\ \mu m^3$ [37] and the average density of 1 g ml$^{-1}$ which results in $8.5 \pm 0.38 \times 10^{-13}$ g cell$^{-1}$. The numerical solution of equation (3.3) for various initial values of $G_p$ is shown in figure 4. It can be seen that there is periodic retardation in growth.

Figure 4 also displays the growth curve in the hypothetical case where $G_p$ and $A$ subsisted at steady state (black curve).

The initial value of $G_p$ for the growth with constant growth rate (black monotonic curve) was chosen such that biomass is comparable to that for oscillating growth in the first 10 h. If the same initial values as for the growth with varying growth rate were chosen, biomass would grow to higher values right from the beginning. Thus, the numerical calculations suggest that oscillating growth for this system is not in favour of increasing growth rate. As can be seen from electronic supplementary material, figure S5, the steady-state growth rate overtakes the oscillating growth rate at about 10.5 h.

It is an important result that biofilm oscillations can be described by considering a few processes only, which are listed below equation (2.1). They include considerably less processes than the Liu model [4]. However, they do include the diffusion of ammonia to the surroundings, unlike that model. Thus, it is plausible to assume that these are the most relevant processes for the phenomenon of oscillations.

All other processes (such as the diffusion of glutamate from the interior of the biofilm to its periphery and from there to the environment) can be neglected.

## 3.3. Average concentrations and average growth rate

Motivated by the reasoning in the Introduction, we now compare the average concentrations with the steady-state concentrations. As the model under study is a mixture of a Lotka–Volterra equation, equation (2.1a) and two linear equations (2.1b,c), it can be assumed that the values are equal. To demonstrate this, we divide equation (2.1a) by $G_p$

$$\frac{1}{G_p}\frac{dG_p}{dt} = \frac{d}{dt}(\ln G_p) = k - k_2 A \quad \text{where} \quad k = k_1 G_E - k_4.$$

We integrate over one oscillation period, $T$

$$\int_0^T \frac{d}{dt}(\ln G_p)\, dt = 0 = \int_0^T (k - k_2 A)\, dt$$

$$= kT - k_2 \int_0^T A\, dt,$$

where the integral is zero because $G_p(T) = G_p(0)$

$$\frac{1}{T}\int_0^T A\, dt = \langle A \rangle = \frac{k}{k_2} = A_{ss}.$$

Now, we calculate the integral of $dA/dt$:

$$\int_0^T \frac{dA}{dt}\, dt = 0 = \int_0^T (k_5 G_i - k_3 A)dt$$

$$= k_5 \int_0^T G_i\, dt - k_3 \int_0^T A\, dt$$

$$k_5 \langle G_i \rangle = k_3 A_{ss} = \frac{k\, k_3}{k_2}$$

$$\langle G_i \rangle = \frac{k\, k_3}{k_2 k_5} = G_{i,ss}.$$

Now, we calculate the integral of $dG_i/dt$ and derive, analogously

$$\langle G_p \rangle = G_{p,ss}.$$

Thus, the average concentrations are equal to the steady-state concentrations.

The question arises whether the oscillations have an effect on the average of the bilinear term $k_2 A G_p$. This is not immediately clear, although figure 4 suggests that the average growth rate is slower than that at the metabolic steady state. Note that the growth term $bAG_pB$ is trilinear.

To check whether $\langle A\, G_p \rangle = A_{ss}.\, G_{p,ss}$, we integrate $dG_p/dt$ over one period, $T$

$$\int_0^T \frac{dG_p}{dt}\, dt = 0 = \int_0^T k G_p dt - \int_0^T k_2 A G_p dt$$

$$= k\langle G_p \rangle T - k_2 A\, G_p T.$$

Since $\langle G_p \rangle = G_{p,ss}$ and $\langle A \rangle = A_{ss} = \dfrac{k}{k_2}$, dividing by $k_2$ and $T$ gives

$$\langle A\, G_p \rangle = A_{ss}.\, G_{p,ss}.$$

Thus, the average of the bilinear term, which can be interpreted as the input to biomass, is indeed unaffected by oscillations, although the ammonia and peripheral glutamate levels oscillate asynchronously. For two-dimensional Lotka–Volterra systems, this property was shown earlier [24].

## 3.4. Bifurcations

Figure 5 shows the two bifurcations: the transcritical bifurcation occurring at $G_E = 5.88$ mmol l$^{-1}$ and the Hopf bifurcation at $G_E = 24.41$ mmol l$^{-1}$, i.e. the transition from stable steady state to stable limit cycle.

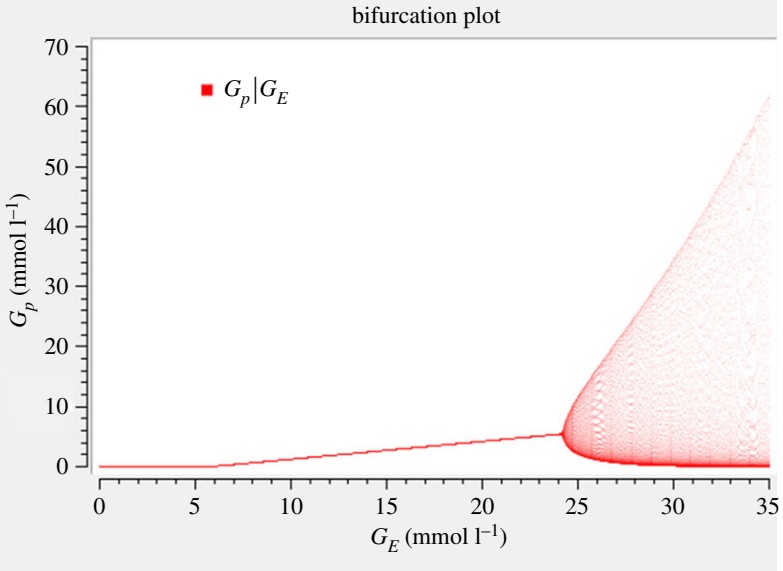

**Figure 5.** Bifurcation diagram of $G_p$ versus $G_E$. For parameter values except $G_E$, see table 1. The transcritical bifurcation occurs at $G_E = 5.88$ mmol $l^{-1}$ and the supercritical Hopf bifurcation at $G_E = 24.41$ mmol $l^{-1}$. The two arms of the convex hull represent the amplitude of oscillation which widen with increasing value of $G_E$. It can be seen that the model is sensitive to $G_E$ in terms of oscillation amplitude.

These values can also be calculated from the general formulae for the bifurcations [19]. The steady-state value of $G_p$ is a linear function of the bifurcation parameter $G_E$, as shown in equation (3.2a). It can be seen that the Hopf bifurcation is supercritical; that is, the amplitude grows gradually starting from zero and the limit cycle is stable right from the beginning.

Near the Hopf bifurcation, the obtained time course curve (figure 2) is sinusoidal. For $G_E \gg$ 24.41 mmol $l^{-1}$ the oscillations get spike-like and are no longer sinusoidal. It is of interest to speculate about the physiological advantage of spike-like oscillations. This question has been discussed earlier in the context of calcium oscillations [26,38,39]. Whenever the kinetic effect of the oscillating variable (e.g. in activating a protein or in a biochemical conversion) is nonlinear and follows a convex function, the spikes contribute more than proportionately to the effect. Thus, spike-like oscillations can lead to a high average effect even at low average value of the variable. In order that oscillations really enable division of labour in the case of biofilms, it can be expected that they should not be sinusoidal. This deserves further studies. A biological explanation of the bifurcations is given in the Discussion.

We checked the parameter sensitivity in two ways. First, we performed a bifurcation analysis for all parameters (except $k_1$ since it is equivalent to $G_E$), see figures 5 and 6 and electronic supplementary material, figures S1–S3. We see a steep increase in the oscillation amplitude with respect to $G_E$ and $k_3$, a moderately steep increase for $k_4$, whereas the other parameters show a very gradual increase in the vicinity of the bifurcation. Second, we applied local parameter sensitivity analysis to the steady-state concentrations, which are equal to the average values. This can be done in an analytical way by differentiating the steady-state values given in equation (3.2) consecutively with respect to all parameters. The resulting unscaled sensitivities for the parameter values from table 1 are given in electronic supplementary material, table S1a. Thereafter, we computed the scaled sensitivities by multiplying by the parameter and dividing by the concentration (electronic supplementary material, table S1b). The obtained values for all sensitivities were confirmed by numerical computation using COPASI [40].

The results show that $A$ is not sensitive at all to $k_3$ and $k_5$, nor is $G_p$ to $k_5$. This is counterintuitive because increasing $k_5$ corresponds to over-expression of glutamate dehydrogenase, which produces ammonia. However, in our model, increasing $k_5$ leads to a decrease in $G_i$, so that the term $k_5 G_i$ stays constant. It deserves further study whether a more realistic kinetics leads to non-zero sensitivity for these cases.

Scaled sensitivities are equal to unity if the parameter enters the formula for the steady value in a multiplicative way (that is, as a factor in the numerator) and equal to minus unity if it is a factor in the denominator. Examples are provided by the scaled sensitivities of $G_i$ and $G_p$ with respect to $k_3$ and of all concentrations with respect to $k_2$, respectively. The highest scaled sensitivity, 1.24, is found

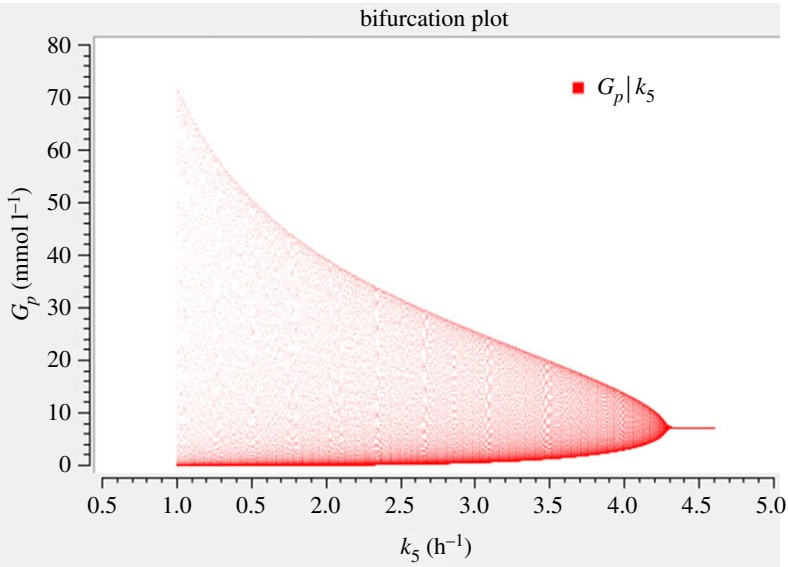

**Figure 6.** Bifurcation diagram of $G_p$ versus $k_5$. For parameter values, see table 1, except for $k_5$, which is varied between 1 and 4.6 $h^{-1}$. It can be seen that the model is less sensitive to $k_5$ in terms of oscillation amplitude when compared with $G_E$.

for all concentrations with respect to $k_1$. That value means that a 1% increase in a parameter value leads to a 1.24% increase in the concentration.

## 3.5. Quasi-steady-state approximation

To ascertain the cause of oscillations, which could be a negative feedback with a delay or a positive feedback, we can study a subsystem by eliminating a variable. This can be done by the quasi-steady-state approximation (QSSA) [15]. In our system, we see that $G_p$ exerts a positive feedback on itself which is linear and, thus, quite weak. For example, in the Higgins–Selkov oscillator involving two variables, the feedback is quadratic [16,17]. Moreover, the above system involves a negative feedback: $G_p$ is converted to $G_i$, $G_i$ is converted to $A$ and $A$ promotes the degradation of $G_p$ (figure 1). In the Goodwin oscillator, which also consists of three variables, a negative feedback is the cause of oscillation [41,42].

Inspired by the observation in figure 5 that the oscillations vanish at high $k_5$ values, we apply the QSSA for $G_i$. This corresponds to the special case where glutamate dehydrogenase is overexpressed. In that case, indeed quenching of oscillations was observed in experiment and also predicted by the Liu model [4]. The variable $G_i$ then attains a quasi-steady state

$$\frac{\mathrm{d}G_i}{\mathrm{d}t} = k_4 G_p - k_5 G_i = 0. \tag{3.4}$$

This leads to

$$G_i = \frac{k_4}{k_5} G_p. \tag{3.5}$$

Substituting this into the above model equations (2.1a–c) yields a simplified system

$$\frac{\mathrm{d}G_p}{\mathrm{d}t} = (k_1 G_E - k_4)G_p - k_2 G_p A \tag{3.6a}$$

and

$$\frac{\mathrm{d}A}{\mathrm{d}t} = -k_3 A + k_4 G_p. \tag{3.6b}$$

It is of interest to analyse its dynamics. It follows from the general result by Hanusse [43,44] that it cannot give rise to a limit cycle because it involves two variables and only linear and bilinear terms. However, the question still remains whether it gives rise to a stable or unstable steady state, whether damped oscillations are possible, etc.

System (3.6) shows two steady states

$$G_{p1} = A_1 = 0, \tag{3.7a,b}$$

which is the TSS, and

$$G_{p2} = \frac{k_1 G_E - k_4}{k_2 k_4} k_3, \quad A_2 = \frac{k_1 G_E - k_4}{k_2}, \tag{3.8a,b}$$

which is the NTSS (see equations (3.2a,b)). The Jacobian matrix reads

$$\mathbf{M} = \begin{pmatrix} k_1 G_E - k_4 - k_2 A & -k_2 G_p \\ k_4 & -k_3 \end{pmatrix}, \tag{3.9}$$

while for the TSS, it reads

$$\mathbf{M} = \begin{pmatrix} k_1 G_E - k_4 & 0 \\ k_4 & -k_3 \end{pmatrix}. \tag{3.10}$$

For matrices with such a triangular structure, the eigenvalues are given by the diagonal elements. In our case

$$\lambda_1 = k_1 G_E - k_4, \quad \lambda_2 = -k_3. \tag{3.11}$$

In any case, the eigenvalues are real, so that not even damped oscillations are possible. For $k_1 G_E < k_4$, both eigenvalues are negative, so that the TSS is a stable node. For $k_1 G_E > k_4$, one eigenvalue is negative and the other one positive. The steady state then is unstable, it is a saddle point.

For the NTSS (3.8), the Jacobian matrix becomes

$$\mathbf{M} = \begin{pmatrix} 0 & -\dfrac{k_1 G_E - k_4}{k_4} k_3 \\ k_4 & -k_3 \end{pmatrix}. \tag{3.12}$$

The characteristic equation reads

$$\lambda^2 + k_3 \lambda + (k_1 G_E - k_4) k_3 = 0 . \tag{3.13}$$

This has the solutions

$$\lambda_{1/2} = -\frac{k_3}{2} \pm \sqrt{\frac{k_3^2}{4} - (k_1 G_E - k_4) k_3}. \tag{3.14}$$

Now, we distinguish three cases:

(a) For $k_1 G_E < k_4$, the term under the square root is positive, so that the root is real. Moreover, it is larger than $k_3/2$. Thus, one eigenvalue is negative and the other one positive. The steady state then is unstable, it is a saddle point.
(b) For $0 < k_1 G_E - k_4 < k_3/4$, the root is again real. It is less than $k_5/2$, though. Both eigenvalues are negative; the steady state is a stable node.
(c) For $k_1 G_E - k_4 > k_3/4$, the root is imaginary. Both eigenvalues are complex numbers, with the same negative real part $-k_3/2$. The steady state is a stable focus. This state is, thus, reached by damped oscillations.

From these calculations, the following conclusions can be drawn. At $k_1 G_E = k_4$, the two steady states of the simplified system (3.6) coincide, as in the complete system (2.1). Since the TSS and NTSS interchange their stability at that point, it is a transcritical bifurcation.

There is a second transition point where the qualitative behaviour changes, at $k_1 G_E - k_4 = k_3/4$. This is another point than the Hopf bifurcation in the complete system, which is at $k_1 G_E - k_4 = k_3 + k_5$. At this transition in the simplified system, a stable node turns into a stable focus. Such a transition must also occur in the complete system between the transcritical bifurcation, where an unstable node turns into a stable node, and the Hopf bifurcation, where a stable focus gets unstable. It is difficult to find it exactly in the three-dimensional system. Beyond this transition, the simplified system shows damped oscillations. This implies that the positive feedback of $G_p$ on itself can be considered as a cause of oscillation, yet not as a cause of a limit cycle. The 'inflation' of the oscillation to a limit cycle in the

complete system appears to be brought about by the negative feedback via $G_i$. The loop via $G_i$ can, moreover, be interpreted as a delay.

In this subsection, we have considered the special case of high $k_5$. This corresponds to a situation realized in experiments by Liu *et al.* where they overexpress the glutamate dehydrogenase leading to an excessive production of ammonia [4]. In that situation, indeed no oscillations were observed. We have proved analytically that the limit cycle disappears in that case. This generalizes the numerical finding shown in figure 6. Thus, to model limit cycle oscillations in biofilms by equation (2.1), we need the full three-dimensional system with values of $k_5$ that are not too high.

The QSSA for the rate constant $k_3$ is given in the electronic supplementary material.

## 4. Discussion

Here, we have used the smallest chemical system showing a Hopf bifurcation to model metabolic oscillations in *B. subtilis* biofilms. That model had been used earlier to describe p53 oscillations [22]. Here, we have applied the model to describe another biological phenomenon. We have specifically selected the parameter values to describe biofilm dynamics, which makes the model more relevant in the light of the biological observation. In our system, the diffusion of ammonia is critical for biofilm oscillations. All the terms in the model are linear, except $k_2 A G_p$, which is bilinear. The model describes metabolic and diffusion processes as outlined above. As an output, the growth of the biofilm (consisting of incremental and halting phases) was also computed (figure 4).

A major reason of the observed oscillations was demonstrated to be the division of labour between the central and peripheral zones of the biofilm. While the release of usable ammonia is mainly delimited to the former, the production of biomass and, thus, growth, is mainly delimited to the latter.

We have presented bifurcation diagrams, which clearly show supercritical Hopf bifurcations (figure 5 and 6, electronic supplementary material, figures S1–S3). Earlier, Wilhelm & Heinrich [19,20] had analysed that bifurcation and had presented one bifurcation diagram. Here, we have added some mathematical analysis. For example, we show the maxima and minima of oscillations, the knowledge of which gives us a quantitative insight into the biofilm dynamics. Moreover, we performed a QSSA and probed for the equality property of the average values. We analysed the Hopf bifurcation by changing not only the external glutamate $G_E$ but alternatively also all the rate constants except $k_1$, since changing $k_1$ has the same effect as changing $G_E$. Interestingly, a recent model [7] has shown a subcritical bifurcation in describing the behaviour of the stress levels in the biofilm periphery. However, they modelled the stress with a single delay differential equation and did not consider other molecular details, while we do not consider stress. Using a single differential equation meets the quest for minimality. However, a delay differential equation (meaning that the time derivative of a variable depends on that variable at a previous time point) is, from a mathematical point of view, very complex because it requires infinitely many initial values (from zero to the delay period, with a simplifying assumption being that they are all equal). Moreover, stability analysis is then considerably more complicated. Our model is complementary to their model. It is closer to Liu's original model [4] but much simpler because it involves only three rather than six variables, and thus only requires three initial values. We chose the parameters of the model such that they are in agreement with Liu's experimental results, namely the period and the amplitude of oscillations.

In our model, peripheral glutamate exerts a positive feedback on itself. Mathematically, this has the form of a bilinear term involving peripheral and external glutamate concentrations. At very low values of $G_E$, the feedback is not strong enough to enable a positive steady state. The system then tends to the TSS. In that state, $G_p$ is zero, so that growth is impossible. Biologically, this can be interpreted that the biofilm is too small to be viable. This is in agreement with observations in the recent study from the Suel group [7]. At a certain threshold value of $G_E$ (5.88 mmol l$^{-1}$), the NTSS turns stable in a transcritical bifurcation. Beyond that value, the feedback is strong enough to enable growth. At high values of $G_E$, the feedback becomes so strong that an overshoot occurs: more glutamate is taken up than needed, so that the $G_p$ level transiently exceeds the steady-state value. Then, more peripheral glutamate is consumed for release of ammonia or for growth, so that the concentration decreases again. This leads to oscillations.

From a functional point of view, a steady state is quite appropriate [29]. Growth of the biofilm does not require oscillations. However, in this system, oscillations help in mitigating the chemical attack that challenges the biofilm [4]. This may have interesting clinical implications in view of treatment of biofilm-forming bacteria by antibiotics. Furthermore, another study [9] indicates that oscillations in growth

actually help in sharing the nutrients among several biofilms more efficiently. However, not all biofilms show oscillations, indicating that it is not critical for biofilms. Our numerical calculations suggest that the average growth rate is lower when compared with growth at the metabolic steady state.

By contrast, the individual concentration variables (ammonia, peripheral glutamate etc. but not biomass) show the equality property that their average during oscillations equals the value at the unstable state, as usual for Lotka–Volterra systems [25]. Here, we have shown that the bilinear input term $k_2 A G_p$ exhibits this equality property as well. This may come as a surprise because the ammonia and peripheral glutamate levels oscillate with a phase shift.

In the paper by Liu *et al.* [4], the oscillations computed by their model have a sinusoidal shape. In our model, such a shape only occurs in the neighbourhood of the Hopf bifurcation. Further beyond it, the shape is more spike-like with the crests being sharper than the troughs.

The question arises whether the model used and analysed here is minimal. On the basis of ODE systems (without delays), at least two variables are needed to generate oscillations [13,14]. However, when only linear and bilinear terms are included, at least three variables are needed, as was proved by Hanusse by an analysis of the Jacobian matrix [43,44]. As shown by Wilhelm & Heinrich [19], such a model requires at least five reactions. Thus, the above model is minimal in terms of the number of variables (criterion with highest priority) and number of reactions, given the type of kinetics. The famous two-variable Brusselator model [45] and the Higgins–Selkov oscillator involve a term of degree three each [16,17]. If the number of reactions is granted the highest priority, the model may look different. Thus, it depends on the criteria what a minimal model is. Note that a delay differential equation [7] is, from the viewpoint of the number of initial values, of infinite dimension. While our model is not necessarily the simplest, it provides a trade-off between simplicity and adequacy to match the observed oscillation in biofilms.

As for any oscillatory system, it is interesting to elucidate the feedback structure. The term $k_1 G_E G_p$ represents a positive feedback because peripheral glutamate stimulates its own uptake. This is because glutamate is a proxy for the concentration of various transport proteins embedded in the cell membranes. The higher the concentration of these proteins, the higher is the glutamate uptake rate. Since this positive feedback is the driving force for oscillations, at low values of $k_1 G_E$, we observe a steady state rather than oscillations. In glycolytic and calcium oscillations, the cause of oscillations is also a positive feedback [13,14,17,46], while in a Goodwin oscillator, it is a negative feedback [41,42].

In addition to the positive feedback, there is also a negative feedback loop in the system (figure 1). As seen from the differential equations, peripheral glutamate positively influences internal glutamate, which positively affects ammonia, which then negatively influences peripheral glutamate. Thus, the overall effect is inhibitory. This feedback structure of the Wilhelm–Heinrich model has been highlighted earlier [22].

By applying the QSSA, we have proved analytically that the limit cycle disappears if glutamate is degraded very fast or ammonia diffuses very easily. As mentioned in the Results section, the former case corresponds to a situation realized in experiments by overexpressing the glutamate dehydrogenase [4]. In that situation, no oscillations were observed. By contrast, in the case where oscillations occur, a description by a simple mass-action system requires three variables. Analyses in this direction may be relevant for clinical interventions via inhibition or activation of bacterial enzymes or changing diffusivity in the biofilm.

The model analysed here has several pros and cons. In view of the mathematical analysis, its simplicity is certainly an advantage. In view of an adequate description of the biological and biochemical processes involved, the model may appear oversimplified. For example, describing growth by trilinear term is quite simplistic; usually, it is described by saturation kinetics (e.g. Michaelis–Menten kinetics). In addition, the assumption that glutamate uptake by the periphery is proportional to the glutamate concentration as explained above could be refined in future studies. Moreover, diffusion processes are usually reversible. In the above model, we neglected the backward processes in diffusion, which is justified if concentration differences are high.

Many theoretical and experimental studies have been published on glycolytic oscillations [13,16,17,47,48]. However, these oscillations only occur under very special or even artificial conditions. In living cells, metabolic oscillations are rare, while being quite frequent in signalling systems [13,14,49,50]. The lights of a car are a helpful analogy: the headlights illuminate the street in a permanent way; there is no point in oscillations. By contrast, the side indicators (as signalling device) flash; that is, they emit oscillating light. Interestingly, in the case of biofilms, metabolic oscillations could provide advantages. While the work described here is quite theoretical, we consider it to be an appropriate basis for refined and more sophisticated models of biofilm oscillations.

Data accessibility. No custom code and datasets was used within this study.

Authors' contributions. S.S. conceived the idea of applying the Wilhelm and Heinrich model to the biofilm system. R.G. performed modelling and simulation using COPASI. He and Á.T.K. interpreted the model variables, parameters and results biologically. B.I. verified the results using Matlab and also produced figures and mathematical analyses. S.S. performed QSSA and mathematical proof and mathematical interpretation of the results. R.G. and S.S. wrote the manuscript and B.I. and Á.T.K. edited and structured it. All the authors have read and approved the manuscript.

Competing interests. The authors declare no competing financial interests.

Funding. R.G. was supported by the Max Planck Society through the IMPRS 'Exploration of Ecological Interactions with Molecular and Chemical Techniques'. B.I. was supported by the DFG through the Collaborative Research Center 1127, ChemBioSys Project C07.

Acknowledgements. We acknowledge David Heckel for his valuable inputs in interpreting the biological implications of the results and Frank Hilker for drawing our attention to the paper by Goel et al. [24].

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
