## [Reviewer comments · Royal Society Open Science]

Review History

RSOS-190810.R0 (Original submission)

Review form: Reviewer 1

Is the manuscript scientifically sound in its present form?

Yes

Are the interpretations and conclusions justified by the results?

Yes

Is the language acceptable?

Yes

Do you have any ethical concerns with this paper?

No

Have you any concerns about statistical analyses in this paper?

No

Recommendation?

Accept with minor revision (please list in comments)

Comments to the Author(s)

The authors successfully applied a previously published model (Wilhelm and Heinrich, 1995) to the oscillatory dynamics of glutamate and ammonia in biofilms. The inspiration comes from the experimental discovery of the oscillatory exchange of the metabolites in microbial (*Bacillus subtilis*) biofilm communities (Liu et al., 2015). It should be emphasised that Liu et al. have already developed a mathematical model for the discovered oscillatory biofilm dynamics; however, the novelty of the present manuscript is that the authors propose a reduced, more simplistic, the so-called minimal model for the oscillatory dynamics of glutamate and ammonia in biofilms. This is an important contribution, the minimal models are of particular importance, not only from the mathematical point of view, but also biologically because of extracting and emphasising the key cellular/molecular mechanisms responsible for the oscillatory behaviour. The presentation of the model is equipped with a comprehensive analysis of the stationary and oscillatory regimes, as well as the transitions from the non-oscillatory to oscillatory regimes via Hopf bifurcations.

I do appreciate the authors' contribution; the proposed cellular mechanisms incorporated in the proposed mathematical model, and the corresponding model predictions, improve the current understanding of the biofilm dynamics, and I do recommend publishing this manuscript after considering the comments listed below.

Major Comments

Comment 1

In the Results, the mathematical outcomes of the model are presented in detail, and the way of the presentation is following the mathematical formalism; however, there is a lack of results with biological importance, or at least their biological interpretation could be enhanced, and I would recommend emphasising the biological value of the results. The authors mention some biological relevant results in the Discussion. These results could be moved to the section Results and presented in more detail; for example, the part where the authors briefly mention how the quasi-steady-state approximation could be applied to an extreme case of very fast diffusion, and that this case resembles the experiments by Liu et al. (2015) where an overexpression of glutamate dehydrogenase leads to an excessive production of ammonia. In the paper of Liu et al. (2015) this finding is much more emphasised and its clinical value is discussed, since it is promising for potential destructions of harmful bacterial biofilms. Therefore, I would recommend to move this part of Discussion into Results, whereas in the Discussion its biological and clinical importance could still be discussed. It would be interesting to see if the newly proposed minimal model is able to provide qualitatively, or even quantitatively, the same results as the previous model by Lie et al., where they showed that the model fits well the experimental data from the cells overexpressing the glutamate dehydrogenase (see Fig. 4 in Lie et al. (2015)).

Comment 2

In the Discussion, the authors write that the oscillations in the model by Liu et al. (2015) have a sinusoidal shape, whereas in the present model the shape is more spike-like, and that the sinusoidal form of the oscillations can only be observed in the proximity of the Hopf bifurcations. It is not completely clear what the authors would like to emphasise here. The authors should evaluate and further discuss this observation, e.g., is the spike-like shape of the oscillations physiologically more relevant for the system under consideration? A relevant evaluation could be made with a direct comparison of the model predictions with the experimental data. Moreover, it would also be of interest to put this observation into the context of the authors' further discussions, at the end of the section (p. 19), where the authors polemize the oscillations in metabolic systems. If the metabolic oscillations are rare in living cells, as the authors claim, and with an impression of generally smoother behaviour of metabolic processes a question arises: is

there of any advantage to be the oscillations in this model sharper, having a spike-like shape? This spike-like form is mostly a hallmark of the oscillations in signalling systems.

Minor Comments

Comment 3

P. 16, line 52: The comparison with the original model by Wilhelm and Heinrich (1995) gives an impression that the only difference in the bifurcation diagram represents the added minima and maxima. The authors should emphasise also other novelties, i.e., the important differences in the new biological meaning of the results. For example, the specifically selected parameter values for biofilm dynamics represent an added value, and the authors could also discuss the biological meaning of the minima/maxima of their functional dependency on the bifurcation parameter. Is the change in amplitudes of the oscillations physiological relevant?

Comment 4

P. 17, line 8: There is a typo: "...are costly to produce and are therefore they are constant."

Review form: Reviewer 2

Is the manuscript scientifically sound in its present form?

No

Are the interpretations and conclusions justified by the results?

No

Is the language acceptable?

Yes

Do you have any ethical concerns with this paper?

No

Have you any concerns about statistical analyses in this paper?

No

Recommendation?

Major revision is needed (please make suggestions in comments)

Comments to the Author(s)

In Garde et al., the authors use a minimal mathematical model to explain the metabolic oscillations observed in *Bacillus subtilis* biofilm which were reported by Liu et al. This minimal model was originally proposed by Wilhelm and Heinrich and has been used in other oscillating systems, for example, to explain p53 oscillations. Liu et al.'s work already has a mathematical model, that was capable to reproduce the experimental observations, as well as to predict the outcome of several perturbations. Interestingly, the Wilhelm and Heinrich model only depends on one non-linear term (K2AGp), which makes this model a good candidate to simplify the conceptual framework to obtain the oscillatory behavior of biofilm metabolism under stress.

Major comments:

1. In the equation for glutamate on the periphery (GP) there is the term K1GEGP which contains self-amplification of glutamate. Does the model oscillate without considering self-amplification of glutamate?

2. Related to the previous question, what is the reason to restrict self-amplification of glutamate to cells located on the periphery and not on the interior? If the reason is because of their differences in their metabolic state, the authors need to consider metabolic state in the equation (more realistic model) and demonstrate that it can be reduced to consider just self-amplification on the periphery.

3. Values used for K_4 and K_3 should not be equal. Glutamate is an amino acid and ammonia is a gas, then these two components cannot diffuse with the same rate. To validate this model it is necessary to find another set of values with biological meaning.

4. How restricted are the parameters of the model to obtain oscillations? The authors could present a sensitivity analysis to study how much it varies.

5. From the study of the model, the authors conclude that oscillating growth is unfavorable for this system (page 9, and conclusion). This conclusion contradicts the original article where they found that oscillations, under low concentrations of glutamate, mitigate metabolic stress. Then, I don't understand their point of view on metabolic oscillations. It is necessary to clarify this point.

6. There is no explanation for K_6 .

7. In the conclusions of Garde et al., they explain that Liu et al. model is not accurate biologically because the amount of these compounds cannot fluctuate. But Liu et al. considered GDH enzyme and ribosomal activities as variables of the system. They were not considering the amount of GDH enzyme and ribosomes in the cells. For this reason, this critic is not reasonable.

With the application of Wilhelm and Heinrich model in metabolic oscillations in biofilms, the authors did not contribute with new insights, nor improving new understanding of the studied system. There is missing biological insights into the different assumptions on which the model is based. However, I think it is a valid exercise to apply a known simpler model to describe oscillations in metabolism. For this reason, I find the manuscript of some value but would suggest a major revision of this paper. Also, a major revision of the text is necessary. Some examples of issues that need to be revised:

- There are missing references in several statements
- Avoid informal language, for example, 'a lot' and subjective words, such as 'nicely'
- Avoid unnecessary references to other mathematical models on the text

Minor comments:

- The authors mentioned that they tried 'several published models of oscillating systems' but they don't mention the obtained results and why they rejected this alternative models.
- Liu et al. considered housekeeping proteins, which can be considered as ribosomal activity. I would specify this in the text, instead of saying that they considered 'ribosomes'.
- Refer reaction K_5 as 'reaction 5'.
- Typos on page 7 (mMoll), page 11 (3a), x-label units in figure 3, and page 16 ('have analysed' should be analysed).
- There is no description of acronym NTSS (Non-trivial steady state) in Figure 3.

Review form: Reviewer 3

Is the manuscript scientifically sound in its present form?

Yes

Are the interpretations and conclusions justified by the results?

Yes

Is the language acceptable?

Yes

Do you have any ethical concerns with this paper?

No

Have you any concerns about statistical analyses in this paper?

No

Recommendation?

Major revision is needed (please make suggestions in comments)

Comments to the Author(s)

Please see attached file (Appendix A).

Decision letter (RSOS-190810.R0)

16-Sep-2019

Dear Dr Ibrahim,

The editors assigned to your paper ("Metabolic oscillations in *Bacillus subtilis* biofilms can be described by a minimal mathematical model") have now received comments from reviewers. We would like you to revise your paper in accordance with the referee and Associate Editor suggestions which can be found below (not including confidential reports to the Editor). Please note this decision does not guarantee eventual acceptance.

Please submit a copy of your revised paper before 09-Oct-2019. Please note that the revision deadline will expire at 00.00am on this date. If we do not hear from you within this time then it will be assumed that the paper has been withdrawn. In exceptional circumstances, extensions may be possible if agreed with the Editorial Office in advance. We do not allow multiple rounds of revision so we urge you to make every effort to fully address all of the comments at this stage. If deemed necessary by the Editors, your manuscript will be sent back to one or more of the original reviewers for assessment. If the original reviewers are not available, we may invite new reviewers.

- Data accessibility

If you wish to submit your supporting data or code to Dryad (<http://datadryad.org/>), or modify your current submission to dryad, please use the following link:
<http://datadryad.org/submit?journalID=RSOS&manu=RSOS-190810>

- Competing interests

- Authors' contributions

- Acknowledgements

- Funding statement

Kind regards,

Andrew Dunn

on behalf of Dr Jose Carrillo (Associate Editor) and Mark Chaplain (Subject Editor)
openscience@royalsociety.org

Comments to Author:

Reviewers' Comments to Author:

Reviewer: 1

Comments to the Author(s)

The authors successfully applied a previously published model (Wilhelm and Heinrich, 1995) to the oscillatory dynamics of glutamate and ammonia in biofilms. The inspiration comes from the experimental discovery of the oscillatory exchange of the metabolites in microbial (*Bacillus subtilis*) biofilm communities (Liu et al., 2015). It should be emphasised that Liu et al. have already developed a mathematical model for the discovered oscillatory biofilm dynamics; however, the novelty of the present manuscript is that the authors propose a reduced, more simplistic, the so-called minimal model for the oscillatory dynamics of glutamate and ammonia in biofilms. This is an important contribution, the minimal models are of particular importance, not only from the mathematical point of view, but also biologically because of extracting and emphasising the key cellular/molecular mechanisms responsible for the oscillatory behaviour. The presentation of the model is equipped with a comprehensive analysis of the stationary and oscillatory regimes, as well as the transitions from the non-oscillatory to oscillatory regimes via Hopf bifurcations.

I do appreciate the authors' contribution; the proposed cellular mechanisms incorporated in the proposed mathematical model, and the corresponding model predictions, improve the current understanding of the biofilm dynamics, and I do recommend publishing this manuscript after considering the comments listed below.

Major Comments

Comment 1

In the Results, the mathematical outcomes of the model are presented in detail, and the way of the presentation is following the mathematical formalism; however, there is a lack of results with biological importance, or at least their biological interpretation could be enhanced, and I would recommend emphasising the biological value of the results. The authors mention some biological relevant results in the Discussion. These results could be moved to the section Results and presented in more detail; for example, the part where the authors briefly mention how the quasi-steady-state approximation could be applied to an extreme case of very fast diffusion, and that this case resembles the experiments by Liu et al. (2015) where an overexpression of glutamate dehydrogenase leads to an excessive production of ammonia. In the paper of Liu et al. (2015) this finding is much more emphasised and its clinical value is discussed, since it is promising for potential destructions of harmful bacterial biofilms. Therefore, I would recommend to move this part of Discussion into Results, whereas in the Discussion its biological and clinical importance could still be discussed. It would be interesting to see if the newly proposed minimal model is able to provide qualitatively, or even quantitatively, the same results as the previous model by Lie et al., where they showed that the model fits well the experimental data from the cells overexpressing the glutamate dehydrogenase (see Fig. 4 in Lie et al. (2015)).

Comment 2

In the Discussion, the authors write that the oscillations in the model by Liu et al. (2015) have a sinusoidal shape, whereas in the present model the shape is more spike-like, and that the sinusoidal form of the oscillations can only be observed in the proximity of the Hopf bifurcations. It is not completely clear what the authors would like to emphasise here. The authors should evaluate and further discuss this observation, e.g., is the spike-like shape of the oscillations physiologically more relevant for the system under consideration? A relevant evaluation could be made with a direct comparison of the model predictions with the experimental data. Moreover, it would also be of interest to put this observation into the context of the authors' further

discussions, at the end of the section (p. 19), where the authors polemize the oscillations in metabolic systems. If the metabolic oscillations are rare in living cells, as the authors claim, and with an impression of generally smoother behaviour of metabolic processes a question arises: is there of any advantage to be the oscillations in this model sharper, having a spike-like shape? This spike-like form is mostly a hallmark of the oscillations in signalling systems.

Minor Comments

Comment 3

P. 16, line 52: The comparison with the original model by Wilhelm and Heinrich (1995) gives an impression that the only difference in the bifurcation diagram represents the added minima and maxima. The authors should emphasise also other novelties, i.e., the important differences in the new biological meaning of the results. For example, the specifically selected parameter values for biofilm dynamics represent an added value, and the authors could also discuss the biological meaning of the minima/maxima of their functional dependency on the bifurcation parameter. Is the change in amplitudes of the oscillations physiological relevant?

Comment 4

P. 17, line 8: There is a typo: "...are costly to produce and are therefore they are constant."

Reviewer: 2

Comments to the Author(s)

In Garde et al., the authors use a minimal mathematical model to explain the metabolic oscillations observed in *Bacillus subtilis* biofilm which were reported by Liu et al. This minimal model was originally proposed by Wilhelm and Heinrich and has been used in other oscillating systems, for example, to explain p53 oscillations. Liu et al.'s work already has a mathematical model, that was capable to reproduce the experimental observations, as well as to predict the outcome of several perturbations. Interestingly, the Wilhelm and Heinrich model only depends on one non-linear term (K_2AG_p), which makes this model a good candidate to simplify the conceptual framework to obtain the oscillatory behavior of biofilm metabolism under stress.

Major comments:

1. In the equation for glutamate on the periphery (GP) there is the term K_1GEGP which contains self-amplification of glutamate. Does the model oscillate without considering self-amplification of glutamate?
2. Related to the previous question, what is the reason to restrict self-amplification of glutamate to cells located on the periphery and not on the interior? If the reason is because of their differences in their metabolic state, the authors need to consider metabolic state in the equation (more realistic model) and demonstrate that it can be reduced to consider just self-amplification on the periphery.
3. Values used for K_4 and K_3 should not be equal. Glutamate is an amino acid and ammonia is a gas, then these two components cannot diffuse with the same rate. To validate this model it is necessary to find another set of values with biological meaning.
4. How restricted are the parameters of the model to obtain oscillations? The authors could present a sensitivity analysis to study how much it varies.
5. From the study of the model, the authors conclude that oscillating growth is unfavorable for this system (page 9, and conclusion). This conclusion contradicts the original article where they found that oscillations, under low concentrations of glutamate, mitigate metabolic stress. Then, I don't understand their point of view on metabolic oscillations. It is necessary to clarify this point.
6. There is no explanation for K_6 .
7. In the conclusions of Garde et al., they explain that Liu et al. model is not accurate biologically because the amount of these compounds cannot fluctuate. But Liu et al. considered GDH enzyme and ribosomal activities as variables of the system. They were not considering the amount of GDH enzyme and ribosomes in the cells. For this reason, this critic is not reasonable.

With the application of Wilhelm and Heinrich model in metabolic oscillations in biofilms, the authors did not contribute with new insights, nor improving new understanding of the studied system. There is missing biological insights into the different assumptions on which the model is based. However, I think it is a valid exercise to apply a known simpler model to describe oscillations in metabolism. For this reason, I find the manuscript of some value but would suggest a major revision of this paper. Also, a major revision of the text is necessary. Some examples of issues that need to be revised:

- There are missing references in several statements
- Avoid informal language, for example, 'a lot' and subjective words, such as 'nicely'
- Avoid unnecessary references to other mathematical models on the text

Minor comments:

- The authors mentioned that they tried 'several published models of oscillating systems' but they don't mention the obtained results and why they rejected this alternative models.
- Liu et al. considered housekeeping proteins, which can be considered as ribosomal activity. I would specify this in the text, instead of saying that they considered 'ribosomes'.
- Refer reaction K5 as 'reaction 5'.
- Typos on page 7 (mMoll), page 11 (3a), x-label units in figure 3, and page 16 ('have analysed' should be analysed).
- There is no description of acronym NTSS (Non-trivial steady state) in Figure 3.

Reviewer: 3

Comments to the Author(s)

Please see attached file.

Author's Response to Decision Letter for (RSOS-190810.R0)

See Appendix B.

RSOS-190810.R1 (Revision)

Review form: Reviewer 2

Is the manuscript scientifically sound in its present form?

No

Are the interpretations and conclusions justified by the results?

No

Is the language acceptable?

Yes

Do you have any ethical concerns with this paper?

No

Have you any concerns about statistical analyses in this paper?

No

Recommendation?

Major revision is needed (please make suggestions in comments)

Comments to the Author(s)

The revision was easy to follow but the authors didn't address my major concerns.

1. The authors have an asymmetry in the model between cells on the periphery and the interior of the biofilm. The reasoning of this asymmetry is based on differences in the metabolism of both groups. In the text, the authors mention that they assume that adding self-amplification on the interior wouldn't affect but they didn't demonstrate it. Adding this part in the model would allow them to not force manually the restriction of growth to cells on the periphery, being a more realistic model but maintaining the simplicity.

2. In this version of the manuscript, there are two different sets of parameters. Set A with wrong parameter values and set B with reasonable parameter values. The authors mentioned that they used set A to be able to compare their model to Wilhelm and Heinrich's model. I couldn't find this comparison. Having these two sets can confuse readers since set A contains incorrect values. Related to that, on page 8 the authors mentioned that figure 3 was created by using set A but on the figure says they used set B. I think it is necessary to eliminate set A and consider just set B, generating a new figure 3 using these values.

3. Since the presented model contains only six parameters, I suggested performing a sensitivity analysis of the different parameters to see how much the results vary. The authors pointed out that in the first manuscript they already studied the two most relevant parameters. I don't think that this is enough. An example that other parameters could also be relevant is that when they changed the constant rates K3 and K4 they needed to change K1 (not k5 or Ge).

4. I don't think that the authors' interpretation of Liu et al.'s article is correct. Liu et al. don't state biofilm oscillations would always be favorable for increasing growth rate. They were studying a particular case in which there is restriction of nutrients. The final advantage is that by using division of labor under stress, the biofilm protects the interior and the biofilm can be resistant to external attacks, such as chemicals. They didn't mention that oscillations give an advantage in the propagation of the biofilm in all conditions. In my opinion, the current text of the manuscript under revision cannot be accepted for publication.

5. I agree with the authors that in Liu et al. 2015 they were measuring concentration of housekeeping proteins. Nevertheless, housekeeping genes mean that these genes are constitutive, not that they have a constant concentration, especially under stress. Besides, I would like to mention that Liu et al. were considering active glutamate dehydrogenase in their model, not concentration in general (see attached original text). "The concentration of active glutamate dehydrogenase (GDH) in the interior cells (Hi); and the rate of biomass production, which is assumed to be given by the concentrations of housekeeping proteins (such as ribosomal proteins) in the interior (ri) and the periphery (rp)."

Giving these explanations, I hold that Garde et al. cannot make a whole point in their article criticizing the option of Liu et al. about having active GDH and concentration of housekeeping proteins as variables in the model.

6. I agree with reviewer 3 that it would be necessary to use different colors for the different initial values in figure 4, to see the progression of the change. I don't think that it is a valid answer to say that 'Unfortunately this cannot be done in COPASI'.

Taking as an example the previous point, I found that in some cases the answers given by the authors show lack of professionalism. Also, the authors scale up the significance of their findings. I don't think this was a correct choice, since the model didn't provide new biological insights about the system.

Minor points:

- Friedrich Schiller University Jena, Bioinformatics
- T. Kovács proof read the manuscript

Review form: Reviewer 3

Is the manuscript scientifically sound in its present form?

Yes

Are the interpretations and conclusions justified by the results?

Yes

Is the language acceptable?

Yes

Do you have any ethical concerns with this paper?

No

Have you any concerns about statistical analyses in this paper?

No

Recommendation?

Accept with minor revision (please list in comments)

Comments to the Author(s)

The authors have changed and improved the manuscript, with the remarks of Reviewers 1 and 2 often being useful. I was disappointed that many of my remarks were ignored. I suspected that other reviewers would address details of the model and biological interpretations, so I decided to point out opportunities to apply, extend, or compare results of the simple model. However, at nearly every point my remarks were denied or ignored. Indeed, perhaps some of them were beyond the scope of the paper, but why use a simple model that has already been published if you don't want to use it to contend with what's been published? For example, I pointed out that in Martinez-Corral, et al., J. Phil. Trans. R. Soc, 374, 1774 (2019) the original authors introduce their own simplification of the model. In this 2019 paper, they go through a pretty detailed comparison. Surely the authors of this submission could at least compare it? I don't think this request is beyond the scope of the paper at all.

Given that the authors expanded the paper and added interesting discussion inspired by Reviewers 1 & 2, I'm in favor of publishing the paper if the authors address how their model relates to the simplified model published in 2019 by Martinez-Corral, et al.

A minor remark: is it unreasonable to ask that graphs be readable? At multiple points, the authors attribute the impossibility of making graphs more readable to the graphical limitations of COPASI. I don't think this is suitable justification of not making these changes when there are so, so many freely available packages for making excellent, readable graphics. My requests along these lines were things as simple as changing axis labels. Moreover, reading the paper again, I truly think the lack of individual axis labels in the bifurcation plots makes it needlessly difficult to read the figures.

Decision letter (RSOS-190810.R1)

04-Nov-2019

Dear Dr Ibrahim:

Manuscript ID RSOS-190810.R1 entitled "Metabolic oscillations in *Bacillus subtilis* biofilms can be described by a minimal mathematical model" which you submitted to Royal Society Open

Science, has been reviewed. The comments of the reviewer(s) are included at the bottom of this letter.

Unusually, the Editors have recommended a further revision to your paper - as this is rarely granted, and usually only when the Editors and/or reviewers consider your manuscript will be publishable after a second revision, please ensure you fully respond to the critiques of the reviewers: providing a revised paper in track changes version, and also a full point-by-point response to these concerns -- if you do not do so, your paper may be rejected.

Please submit a copy of your revised paper before 27-Nov-2019. Please note that the revision deadline will expire at 00.00am on this date. If we do not hear from you within this time then it will be assumed that the paper has been withdrawn. In exceptional circumstances, extensions may be possible if agreed with the Editorial Office in advance. We do not allow multiple rounds of revision so we urge you to make every effort to fully address all of the comments at this stage. If deemed necessary by the Editors, your manuscript will be sent back to one or more of the original reviewers for assessment. If the original reviewers are not available we may invite new reviewers.

- Ethics statement

- Data accessibility

- Competing interests

- Authors' contributions

- Acknowledgements

- Funding statement

on behalf of Dr Jose Carrillo (Associate Editor) and Mark Chaplain (Subject Editor)
openscience@royalsociety.org

Reviewer comments to Author:

Reviewer: 2

Comments to the Author(s)

The revision was easy to follow but the authors didn't address my major concerns.

1. The authors have an asymmetry in the model between cells on the periphery and the interior of the biofilm. The reasoning of this asymmetry is based on differences in the metabolism of both groups. In the text, the authors mention that they assume that adding self-amplification on the interior wouldn't affect but they didn't demonstrate it. Adding this part in the model would allow them to not force manually the restriction of growth to cells on the periphery, being a more realistic model but maintaining the simplicity.

2. In this version of the manuscript, there are two different sets of parameters. Set A with wrong parameter values and set B with reasonable parameter values. The authors mentioned that they used set A to be able to compare their model to Wilhelm and Heinrich's model. I couldn't find this comparison. Having these two sets can confuse readers since set A contains incorrect values. Related to that, on page 8 the authors mentioned that figure 3 was created by using set A but on the figure says they used set B. I think it is necessary to eliminate set A and consider just set B, generating a new figure 3 using these values.

3. Since the presented model contains only six parameters, I suggested performing a sensitivity analysis of the different parameters to see how much the results vary. The authors pointed out that in the first manuscript they already studied the two most relevant parameters. I don't think that this is enough. An example that other parameters could also be relevant is that when they changed the constant rates K3 and K4 they needed to change K1 (not k5 or Ge).

4. I don't think that the authors' interpretation of Liu et al.'s article is correct. Liu et al. don't state biofilm oscillations would always be favorable for increasing growth rate. They were studying a particular case in which there is restriction of nutrients. The final advantage is that by using division of labor under stress, the biofilm protects the interior and the biofilm can be resistant to external attacks, such as chemicals. They didn't mention that oscillations give an advantage in the propagation of the biofilm in all conditions. In my opinion, the current text of the manuscript under revision cannot be accepted for publication.

5. I agree with the authors that in Liu et al. 2015 they were measuring concentration of housekeeping proteins. Nevertheless, housekeeping genes mean that these genes are constitutive, not that they have a constant concentration, especially under stress. Besides, I would like to mention that Liu et al. were considering active glutamate dehydrogenase in their model, not concentration in general (see attached original text). "The concentration of active glutamate dehydrogenase (GDH) in the interior cells (Hi); and the rate of biomass production, which is assumed to be given by the concentrations of housekeeping proteins (such as ribosomal proteins) in the interior (ri) and the periphery (rp)."

Giving these explanations, I hold that Garde et al. cannot make a whole point in their article criticizing the option of Liu et al. about having active GDH and concentration of housekeeping proteins as variables in the model.

6. I agree with reviewer 3 that it would be necessary to use different colors for the different initial values in figure 4, to see the progression of the change. I don't think that it is a valid answer to say that 'Unfortunately this cannot be done in COPASI'.

Taking as an example the previous point, I found that in some cases the answers given by the authors show lack of professionalism. Also, the authors scale up the significance of their findings. I don't think this was a correct choice, since the model didn't provide new biological insights about the system.

Minor points:

- Friedrich Schiller University Jena, Bioinformatics
- T. Kovács proof read the manuscript

Reviewer: 3

Comments to the Author(s)

The authors have changed and improved the manuscript, with the remarks of Reviewers 1 and 2 often being useful. I was disappointed that many of my remarks were ignored. I suspected that other reviewers would address details of the model and biological interpretations, so I decided to point out opportunities to apply, extend, or compare results of the simple model. However, at nearly every point my remarks were denied or ignored. Indeed, perhaps some of them were beyond the scope of the paper, but why use a simple model that has already been published if you don't want to use it to contend with what's been published? For example, I pointed out that in Martinez-Corral, et al., J. Phil. Trans. R. Soc, 374, 1774 (2019) the original authors introduce their own simplification of the model. In this 2019 paper, they go through a pretty detailed comparison. Surely the authors of this submission could at least compare it? I don't think this request is beyond the scope of the paper at all.

Given that the authors expanded the paper and added interesting discussion inspired by Reviewers 1 & 2, I'm in favor of publishing the paper if the authors address how their model relates to the simplified model published in 2019 by Martinez-Corral, et al.

A minor remark: is it unreasonable to ask that graphs be readable? At multiple points, the authors attribute the impossibility of making graphs more readable to the graphical limitations of COPASI. I don't think this is suitable justification of not making these changes when there are so, so many freely available packages for making excellent, readable graphics. My requests along

these lines were things as simple as changing axis labels. Moreover, reading the paper again, I truly think the lack of individual axis labels in the bifurcation plots makes it needlessly difficult to read the figures.

Author's Response to Decision Letter for (RSOS-190810.R1)

See Appendix C.

RSOS-190810.R2 (Revision)

Review form: Reviewer 2

Is the manuscript scientifically sound in its present form?

No

Are the interpretations and conclusions justified by the results?

Yes

Is the language acceptable?

Yes

Do you have any ethical concerns with this paper?

No

Have you any concerns about statistical analyses in this paper?

No

Recommendation?

Accept with minor revision (please list in comments)

Comments to the Author(s)

I appreciate changes in the new manuscript. I have two comments:

1) Figure 4, the units that the authors used for biomass is mmol/l (mM), and they use it to talk about growth. I would suggest converting it to cells/l since I don't think that mol (6.10^{23}) per liter makes sense, considering the size of cells.

2) Figure 4 and the main text about this part are confusing. The authors state that oscillating growth in biofilms is not in favor of growth rate but in figure 4 it can be appreciated that constant grown (yellow monotonic curve) is slower and doesn't reach the maximum value. In previous versions of the manuscript, the advantage of non-oscillating growth was clearer.

Review form: Reviewer 3

Is the manuscript scientifically sound in its present form?

Yes

Are the interpretations and conclusions justified by the results?

Yes

Is the language acceptable?

Yes

Do you have any ethical concerns with this paper?

No

Have you any concerns about statistical analyses in this paper?

No

Recommendation?

Accept with minor revision (please list in comments)

Comments to the Author(s)

The changes to the manuscript, especially addressing Reviewer 2's comments, have improved the paper.

Am small point I had reading the manuscript this time was the remark "We have chosen the value of the conversion factor b in Eq. (4) such that the doubling time is in agreement with the experimental values[35]" on p7 l14. This is odd because it comes much before Eq. 4. Can you just remove the reference to Eq. 4 here and then reintroduce b when Eq. 4 is defined?

The remaining issue I have is still with the readability of the graphs. In Figure 4, for example, the legend containing B1, B10, etc, while addressing previous reviewer comments, is also rather confusing. The colors them selves are also not on any kind of scale even though the parameter values they represent are. It would be good to have these, for example, as different shades on a continuum with the parameter value defined in color somewhere. I understand you are using COPASI, which apparently has limited graphical capabilities, but is this truly not possible?

Decision letter (RSOS-190810.R2)

02-Jan-2020

Dear Dr Ibrahim:

On behalf of the Editors, I am pleased to inform you that your Manuscript RSOS-190810.R2 entitled "Metabolic oscillations in *Bacillus subtilis* biofilms can be described by a minimal mathematical model" has been accepted for publication in Royal Society Open Science subject to minor revision in accordance with the referee suggestions. Please find the referees' comments at the end of this email.

The reviewers and Subject Editor have recommended publication, but also suggest some minor revisions to your manuscript. Therefore, I invite you to respond to the comments and revise your manuscript.

- Ethics statement

- Data accessibility

If you wish to submit your supporting data or code to Dryad (<http://datadryad.org/>), or modify your current submission to dryad, please use the following link:
<http://datadryad.org/submit?journalID=RSOS&manu=RSOS-190810.R2>

- Competing interests

- Authors' contributions

- Acknowledgements

- Funding statement

Because the schedule for publication is very tight, it is a condition of publication that you submit the revised version of your manuscript before 11-Jan-2020. Please note that the revision deadline will expire at 00.00am on this date. If you do not think you will be able to meet this date please let me know immediately.

To revise your manuscript, log into <https://mc.manuscriptcentral.com/rsos> and enter your Author Centre, where you will find your manuscript title listed under "Manuscripts with

Decisions". Under "Actions," click on "Create a Revision." You will be unable to make your revisions on the originally submitted version of the manuscript. Instead, revise your manuscript and upload a new version through your Author Centre.

on behalf of Dr Jose Carrillo (Associate Editor) and Mark Chaplain (Subject Editor)
openscience@royalsociety.org

Reviewer comments to Author:
 Reviewer: 2

Comments to the Author(s)
 I appreciate changes in the new manuscript. I have two comments:

- 1) Figure 4, the units that the authors used for biomass is mmol/l (mM), and they use it to talk

about growth. I would suggest converting it to cells/l since I don't think that mol (6.10^{23}) per liter makes sense, considering the size of cells.

2) Figure 4 and the main text about this part are confusing. The authors state that oscillating growth in biofilms is not in favor of growth rate but in figure 4 it can be appreciated that constant grown (yellow monotonic curve) is slower and doesn't reach the maximum value. In previous versions of the manuscript, the advantage of non-oscillating growth was clearer.

Reviewer: 3

Comments to the Author(s)

The changes to the manuscript, especially addressing Reviewer 2's comments, have improved the paper.

Am small point I had reading the manuscript this time was the remark "We have chosen the value of the conversion factor b in Eq. (4) such that the doubling time is in agreement with the experimental values[35]" on p7 l14. This is odd because it comes much before Eq. 4. Can you just remove the reference to Eq. 4 here and then reintroduce b when Eq. 4 is defined?

The remaining issue I have is still with the readability of the graphs. In Figure 4, for example, the legend containing B1, B10, etc, while addressing previous reviewer comments, is also rather confusing. The colors them selves are also not on any kind of scale even though the parameter values they represent are. It would be good to have these, for example, as different shades on a continuum with the parameter value defined in color somewhere. I understand you are using COPASI, which apparently has limited graphical capabilities, but is this truly not possible?

Author's Response to Decision Letter for (RSOS-190810.R2)

See Appendix D.

Decision letter (RSOS-190810.R3)

15-Jan-2020

Dear Dr Ibrahim,

It is a pleasure to accept your manuscript entitled "Metabolic oscillations in *Bacillus subtilis* biofilms can be described by a minimal mathematical model" in its current form for publication in Royal Society Open Science. The comments of the reviewer(s) who reviewed your manuscript are included at the foot of this letter.

Due to rapid publication and an extremely tight schedule, if comments are not received, your paper may experience a delay in publication. Royal Society Open Science operates under a continuous publication model. Your article will be published straight into the next open issue and

this will be the final version of the paper. As such, it can be cited immediately by other researchers. As the issue version of your paper will be the only version to be published I would advise you to check your proofs thoroughly as changes cannot be made once the paper is published.

on behalf of Dr Jose Carrillo (Associate Editor) and Mark Chaplain (Subject Editor)
openscience@royalsociety.org

Appendix A

The authors adapt a simple differential equation model to account for the metabolic oscillations in a *Bacillus subtilis* biofilm.

The paper succeeds in presenting a very simple model that accounts for bifurcations to growth and then to oscillations. As stated in the beginning, “Our ultimate aim was to develop a minimal model to describe the metabolic oscillations happening in a biofilm”. The authors have succeeded, however, I’m not sure I would say they developed the model in the paper. In that vein, my biggest criticism is that the paper does not attempt much beyond presenting a simple model, much of which was already done in papers by Wilhelm and Heinrich. There is the quasi-steady state calculation, which does lend some insight. However, it would be interesting if the authors would attempt to see if their model, or some extension of it, could account for some of the other phenotypes observed by Liu and co-authors in various papers. These things should at least be discussed, and more comparisons should be made to subsequent extensions or modifications of the model from Liu, et al. I will discuss a few specific examples below.

While the model in Liu, et al. (2015) has more parameters than the one in the present manuscript, it does account for another observation: the increasing oscillation period that the authors observe over the course of experiments. From the Liu paper: “The model also accounts for the observed slight increase of the oscillation period by considering an increase in the ratio of interior to peripheral cells over time”. Can the simpler model account for such a phenotype? As far as I can tell from the submitted paper, the oscillating regime of the simpler model has a constant period? Either way, this is not discussed in the manuscript. Additionally, the authors of the original Liu paper seem to have presented their own simplified version of their model in Martinez-Corral, et al., *J. Phil. Trans. R. Soc.*, 374, 1774 (2019). How does the model from this paper compare to that simplification? It would be very nice to see a discussion of how the two simplified models compare, and some discussion of the changing period and whether or not it can be achieved in this minimal model. If not, what is the simplest extension of the model that would allow that?

Another phenotype is observed by Liu, et al. is in citation 6 of the submitted manuscript: “Furthermore, another study⁶ indicates that oscillations in growth actually help in sharing the nutrients among several biofilms more efficiently. However, not all biofilms show oscillations, indicating that it is not critical for biofilms.” In the cited paper (6), the authors observe that, in coupled systems, oscillations can go in and out of phase as a function of G_E . Moreover, they claim that coupled oscillations can lead to higher average growth. Could the model presented here be easily applied to that system? It would add more parameters, but if you apply a simple modeling scheme like this one to two biofilms, can you account for these observations by Liu, et al?

Another point that is oddly absent from this manuscript is any discussion of how this model could relate to the discovery by Liu and co-workers that the metabolic oscillations are mediated by ion channel action potentials. While a model that takes that into account is surely outside the scope of this submission, the manuscript could be improved by a thoughtful discussion of how that phenomenon could inform how we think about the simple model presented, and whether some extension could take the action potentials into account. This point seems also related to Martinez-Corral R, et al., *J. Phil. Trans. R. Soc.*, 374, 1774 (2019).

I have one last thought, but I'm not sure if and how the manuscript should be revised in light of this. The title of the paper is "Metabolic oscillations in *Bacillus subtilis* biofilms can be described by a minimal mathematical model". This title, along with a few remarks from the paper (e.g. "not all biofilms show oscillations, indicating that it is not critical for biofilms"), seems to almost suggest that the observed oscillations are trivial or not important to think about. However, this is an entire paper about modeling them? So, it must be worth some effort? Perhaps I'm completely off base here and I'm reading things into the paper that were not intended to be there, but it's a distinct undertone I got from the manuscript. I felt I needed to mention it.

Here are some specific points, some of them quite minor:

Why are Eq. 4a and 4b different equations?

The variable k is defined as different quantities at different points in the paper: p. 9 (note: as indicated on the top of the reviewer copy pages, *not* the bottom of the manuscript), l. 48: $k = k_2 k_6$; p. 11, l. 4: $k = k_1 G_E - k_4$. It seems perhaps they are actually different characters, but it's confusing. Can different notation be used here?

p. 12, l. 31-32: "The steady-state value of G_p is a linear function of the bifurcation parameter G_E , as shown in equation (3b)." I believe this should read "equation (3a)" ?

Some remarks about the figures:

Fig. 2:

- The axis labels are difficult both to read and interpret. I think the y-axis says "mmol/l", but it almost looks like a typo. Perhaps change the label to "concentration (mmol/litre)" or something like that. Same with x-axis: "time (hours)". Also—a small comment, but the main text uses "mM" for millimolar concentration, while the axes of graphs use "mmol/l". Can this be consistent in all cases?
- The legend for $[A]$, $[G_i]$, and $[G_p]$ is so small that it's nearly impossible to read. Please make it larger, and perhaps move it to the top instead of below the x-axis.
- In addition to the graph shown, please include a widened, zoomed in portion, where it's easier to see the relative to phases of $[A]$, $[G_i]$, and $[G_p]$. With the axis so compressed, it's hard to tell.

Fig. 4:

- See Fig. 2 comments about the axes, legends, and labels
- Could the blue curves be color-coded according to different initial values for $[G_p]$? It looks like there is some crossover of the different curves, but it's difficult to tell with all of them being the same color.

Fig. 5:

- See Fig. 2 comments
- Please label the axes individually
- If the line is so thin, please add an extra label pointing out the $G_E = 9.5$ mM bifurcation. It's difficult to see.

Fig. 6

- Please label the axes individually

Appendix B

Manuscript ID RSOS-190810

Title: Metabolic oscillations in *Bacillus subtilis* biofilms can be described by a minimal mathematical model

Answers to the Reviewers' comments

We thank the reviewers for their careful reading of our manuscript and their constructive and helpful comments. Our responses (red color) to the Referee's comments are given below.

Reviewer #1 :

Major Comments

1.1) In the Results, the mathematical outcomes of the model are presented in detail, and the way of the presentation is following the mathematical formalism; however, there is a lack of results with biological importance, or at least their biological interpretation could be enhanced, and I would recommend emphasising the biological value of the results.

We have now inserted a paragraph at the bottom of p.10 saying that an important result is that biofilm oscillations can be described by considering a few processes only, which are listed below Eq. (1).

Moreover, some more biological interpretation has been included as a response to comments 1.2 and 2.5, see below. We feel that especially the latter point is an important biological interpretation, in which we now question (more clearly than before) the claim by other authors that biofilm oscillations would always be favorable for increasing growth rate.

The authors mention some biological relevant results in the Discussion. These results could be moved to the section Results and presented in more detail; for example, the part where the authors briefly mention how the quasi-steady-state approximation could be applied to an extreme case of very fast diffusion, and that this case resembles the experiments by Liu et al. (2015) where an overexpression of glutamate dehydrogenase leads to an excessive production of ammonia.

We have moved the biological interpretation of the case of very fast glutamate dehydrogenase to the Results section (p. 17) and briefly repeat it in the Discussion section. Overall, we feel that interpretation rather belongs to the Discussion section. For example, already in the previous version, we discussed the biological meaning of the transcritical and Hopf bifurcations, and still do so on p. 18.

In the paper of Liu et al. (2015) this finding is much more emphasised and its clinical value is discussed, since it is promising for potential destructions of harmful bacterial biofilms. Therefore, I would recommend to move this part of Discussion into Results, whereas in the Discussion its biological and clinical importance could still be discussed. It would be interesting to see if the newly proposed minimal model is able to provide qualitatively, or even quantitatively, the same results as the previous model by Liu et al., where they showed that the model fits well the experimental data from the cells overexpressing the glutamate dehydrogenase (see Fig. 4 in Liu et al. (2015)).

On p.18, second-last paragraph, and p.20, 1st paragraph, we now mention possible clinical implications for treatment with antibiotics, inhibition/activation of bacterial enzymes or changing diffusivity in the biofilm. However, we feel that a general discussion of clinical implications is premature at this stage of model development.

1.2) In the Discussion, the authors write that the oscillations in the model by Liu et al. (2015) have a sinusoidal shape, whereas in the present model the shape is more spike-like, and that the sinusoidal form of the oscillations can only be observed in the proximity of the Hopf bifurcations. It is not completely clear what the authors would like to emphasise here. The authors should evaluate and further discuss this observation, e.g., is the spike-like shape of the oscillations physiologically more relevant for the system under consideration? A relevant evaluation could be made with a direct comparison of the model predictions with the experimental data. Moreover, it would also be of interest to put this observation into the context of the authors' further discussions, at the end of the section (p. 19), where the authors polemize the oscillations in metabolic systems. If the metabolic oscillations are rare in living cells, as the authors claim, and with an impression of generally smoother behaviour of metabolic processes a question arises: is there of any advantage to be the oscillations in this model sharper, having a spike-like shape? This spike-like form is mostly a hallmark of the oscillations in signalling systems.

It is indeed of interest to speculate about the physiological advantage of spike-like oscillations. We have now included a paragraph on that issue at the bottom of p. 12.

Minor Comments

1.3) P. 16, line 52: The comparison with the original model by Wilhelm and Heinrich (1995) gives an impression that the only difference in the bifurcation diagram represents the added minima and maxima. The authors should emphasise also other novelties, i.e., the important differences in the new biological meaning of the results. For example, the specifically selected parameter values for biofilm dynamics represent an added value, and the authors could also discuss the biological meaning of the minima/maxima of their functional dependency on the bifurcation parameter. Is the change in amplitudes of the oscillations physiological relevant?

We added a sentence on the specifically selected parameter values to the Abstract. Additionally, we added some sentences on the biological meaning of the results in the first and second paragraphs of the Discussion.

1.4) P. 17, line 8: There is a typo: "...are costly to produce and are therefore they are constant."

Done.

Reviewer: 2

2.1) In the equation for glutamate on the periphery (GP) there is the term $K1GEGP$ which contains self-amplification of glutamate. Does the model oscillate without considering self-amplification of glutamate?

At the end of the subsection on model assumptions, we added the following, 'Without this self-amplification of glutamate, the system would not oscillate by construction of the minimal model.'

2.2) Related to the previous question, what is the reason to restrict self-amplification of glutamate to cells located on the periphery and not on the interior? If the reason is because of their differences in their metabolic state, the authors need to consider metabolic state in the equation (more realistic model) and demonstrate that it can be reduced to consider just self-amplification on the periphery.

We have noticed that the paragraphs on the interpretation of the terms and model assumptions were partly redundant. We have combined them into a single subsection on p 3 therein, we added a sentence saying why we neglect the self amplification in G_i .

2.3) Values used for K4 and K3 should not be equal. Glutamate is an amino acid and ammonia is a gas, then these two components cannot diffuse with the same rate. To validate this model it is necessary to find another set of values with biological meaning.

That is a very valid point. We now use two different parameter sets (named A and B, see Table 1). Set A allows a better comparison to the results by Wilhelm and Heinrich¹⁹, while set B is more realistic from a physico-chemical point of view, in particular, w.r.t. the rate constants of diffusion. In set B, we choose a ratio of 2 as it matches the ratio of diffusion coefficients of ammonia to glutamate given in the literature approximately. We explain this at the bottom of p 5.

2.4) How restricted are the parameters of the model to obtain oscillations? The authors could present a sensitivity analysis to study how much it varies.

We had indeed presented sensitivity analysis of two most biologically relevant parameters namely, G_e and k_5 under the subsection Bifurcations.

2.5) From the study of the model, the authors conclude that oscillating growth is unfavorable for this system (page 9, and conclusion). This conclusion contradicts the original article where they found that oscillations, under low concentrations of glutamate, mitigate metabolic stress. Then, I don't understand their point of view on metabolic oscillations. It is necessary to clarify this point.

This is a valid point, therefore we extended the last paragraph on p. 18 (Discussion section), questioning the claim by other authors that biofilm oscillations would always be favorable for increasing growth rate. Moreover, we toned down our statement about whether oscillations would be favourable. We confine it to the effect on growth rate. A thorough comparison of the oscillatory and stationary regimes of biofilms would require a much more detailed study, which is beyond the scope of our paper.

2.6) There is no explanation for K6.

We removed k_6k_2 altogether and replaced it with b .

2.7) In the conclusions of Garde et al., they explain that Liu et al. model is not accurate biologically because the amount of these compounds cannot fluctuate. But Liu et al. considered GDH enzyme and ribosomal activities as variables of the system. They were not considering the amount of GDH enzyme and ribosomes in the cells. For this reason, this critic is not reasonable.

We have rechecked and found that in the supplementary info of their manuscript (2015), Liu et al. mention that these are indeed concentrations, which cannot easily oscillate in our opinion. We corrected the term ribosomes by writing 'housekeeping (e.g. ribosomal) proteins'. We maintain our criticism and explain it better than before in the Intro (p. 2, second-last paragraph) and Discussion (p. 16, second paragraph).

With the application of Wilhelm and Heinrich model in metabolic oscillations in biofilms, the authors did not contribute with new insights, nor improving new understanding of the studied system. There is missing biological insights into the different assumptions on which the model is based. However, I think it is a valid exercise to apply a known simpler model to

describe oscillations in metabolism. For this reason, I find the manuscript of some value but would suggest a major revision of this paper. Also, a major revision of the text is necessary. Some examples of issues that need to be revised:

- There are missing references in several statements

We inserted references at the following places: 3 Refs. on biofilms in the first paragraph of the Intro, one Ref. on the modelling of biofilms at the top of p.3, the monographs “Nonlinear Dynamics and Chaos” by S. Strogatz and “The Regulation of Cellular Systems” by Heinrich and Schuster, three references for diffusion coefficients in table 1.

- Avoid informal language, for example, ‘a lot’ and subjective words, such as ‘nicely’

Done.

- Avoid unnecessary references to other mathematical models on the text

We feel that a comparison with other models is very helpful.

Minor comments:

- The authors mentioned that they tried ‘several published models of oscillating systems’ but they don’t mention the obtained results and why they rejected this alternative models.

We have deleted this sentence because it is unnecessary.

- Liu et al. considered housekeeping proteins, which can be considered as ribosomal activity. I would specify this in the text, instead of saying that they considered ‘Ribosomes’.

See our response to comment 2.7.

- Refer reaction K5 as ‘reaction 5’.

Done.

- Typos on page 7 (mMoll), page 11 (3a), x-label units in figure 3, and page 16 (‘have analysed’ should be analysed).

Done

- There is no description of acronym NTSS (Non-trivial steady state) in Figure 3.

Done

Reviewer: 3

3.1) The authors adapt a simple differential equation model to account for the metabolic oscillations in a *Bacillus subtilis* biofilm.

The paper succeeds in presenting a very simple model that accounts for bifurcations to growth and then to oscillations. As stated in the beginning, “Our ultimate aim was to develop a minimal model to describe the metabolic oscillations happening in a biofilm”. The authors have succeeded, however, I’m not sure I would say they developed the model in the paper. In that vein, my biggest criticism is that the paper does not attempt much beyond presenting a simple model, much of which was already done in papers by Wilhelm and Heinrich. There is the quasi steady state calculation, which does lend some insight. However, it would be interesting if the authors would attempt to see if their model, or some extension of it, could account for some of the other phenotypes observed by Liu and co-authors in various papers. These things should at least be discussed, and more comparisons should be made to subsequent extensions or modifications of the model from Liu, et al. I will discuss a few specific examples below.

While the model in Liu, et al. (2015) has more parameters than the one in the present manuscript, it does account for another observation: the increasing oscillation period that the authors observe over the course of experiments. From the Liu paper: “The model also accounts for the observed slight increase of the oscillation period by considering an increase

in the ratio of interior to peripheral cells over time”. Can the simpler model account for such a phenotype? As far as I can tell from the submitted paper, the oscillating regime of the simpler model has a constant period? Either way, this is not discussed in the manuscript. Additionally, the authors of the original Liu paper seem to have presented their own simplified version of their model in Martinez-Corral, et al., J. Phil. Trans. R. Soc, 374, 1774 (2019). How does the model from this paper compare to that simplification? It would be very nice to see a discussion of how the two simplified models compare, and some discussion of the changing period and whether or not it can be achieved in this minimal model. If not, what is the simplest extension of the model that would allow that?

As we use a minimal model, the above-mentioned phenomenon cannot be described unfortunately. This is an interesting topic for future model extensions.

3.2) Another phenotype is observed by Liu, et al. is in citation 6 of the submitted manuscript: “Furthermore, another study indicates that oscillations in growth actually help in sharing the nutrients among several biofilms more efficiently. However, not all biofilms show oscillations, indicating that it is not critical for biofilms.” In the cited paper (6), the authors observe that, in coupled systems, oscillations can go in and out of phase as a function of GE. Moreover, they claim that coupled oscillations can lead to higher average growth. Could the model presented here be easily applied to that system? It would add more parameters, but if you apply a simple modeling scheme like this one to two biofilms, can you account for these observations by Liu, et al?

Coupled systems are beyond the scope of this manuscript but we have a sequel manuscript about this in the making.

3.3) Another point that is oddly absent from this manuscript is any discussion of how this model could relate to the discovery by Liu and co-workers that the metabolic oscillations are mediated by ion channel action potentials. While a model that takes that into account is surely outside the scope of this submission, the manuscript could be improved by a thoughtful discussion of how that phenomenon could inform how we think about the simple model presented, and whether some extension could take the action potentials into account. This point seems also related to Martinez-Corral R, et al., J. Phil. Trans. R. Soc, 374, 1774 (2019).

This is indeed outside the scope of this manuscript. This was not described by the Liu model (2015) either. We again stress that we have presented a minimal model to describe the basic features of biofilm oscillations.

3.4) I have one last thought, but I’m not sure if and how the manuscript should be revised in light of this. The title of the paper is “Metabolic oscillations in *Bacillus subtilis* biofilms can be described by a minimal mathematical model”. This title, along with a few remarks from the paper (e.g. “not all biofilms show oscillations, indicating that it is not critical for biofilms”), seems to almost suggest that the observed oscillations are trivial or not important to think about. However, this is an entire paper about modeling them? So, it must be worth some effort? Perhaps I’m completely off base here and I’m reading things into the paper that were not intended to be there, but it’s a distinct undertone I got from the manuscript. I felt I needed to mention it.

Our aim was to describe biofilm oscillations that are observed under special conditions. We do not imply that they would be ubiquitous or beneficial.

Here are some specific points, some of them quite minor:

3.5) Why are Eq. 4a and 4b different equations?

We have combined the equations.

3.6) The variable k is defined as different quantities at different points in the paper: p. 9 (note: as indicated on the top of the reviewer copy pages, not the bottom of the manuscript), l. 48: $k = k_2k_6$; p. 11, l. 4: $k = k_1GE - k_4$. It seems perhaps they are actually different characters, but it's confusing. Can different notation be used here?

We have renamed the conversion factor in the biomass equation as b .

3.6) p. 12, l. 31-32: "The steady-state value of G_p is a linear function of the bifurcation parameter GE , as shown in equation (3b)." I believe this should read "equation (3a)" ?

Done.

3.7) Some remarks about the figures:

Fig. 2:

- The axis labels are difficult both to read and interpret. I think the y-axis says "mmol/l", but it almost looks like a typo. Perhaps change the label to "concentration (mmol/litre)" or something like that. Same with x-axis: "time (hours)". Also—a small comment, but the main text uses "mM" for millimolar concentration, while the axes of graphs use "mmol/l". Can this be consistent in all cases?

This is due to the COPASI software and we feel that the meaning is clear. We have also replaced mM with mmol/l throughout the manuscript.

3.8) The legend for $[A]$, $[Gi]$, and $[Gp]$ is so small that it's nearly impossible to read. Please make it larger, and perhaps move it to the top instead of below the x-axis.

We have improved the readability of the figures.

3.9) In addition to the graph shown, please include a widened, zoomed in portion, where it's easier to see the relative phases of $[A]$, $[Gi]$, and $[Gp]$. With the axis so compressed, it's hard to tell.

We solved this by changing the scale of the abscissa. (running the simulations for 25 instead of 50 hours)

3.9) Fig. 4:

- See Fig. 2 comments about the axes, legends, and labels

See above (2.1)

- Could the blue curves be color-coded according to different initial values for $[Gp]$? It looks like there is some crossover of the different curves, but it's difficult to tell with all of them being the same color.

Unfortunately this cannot be done in COPASI.

3.10) Fig. 5:

- See Fig. 2 comments

See above.

- Please label the axes individually

This is due to the format of COPASI. We feel the meaning should be clear from the legend.

- If the line is so thin, please add an extra label pointing out the $GE = 9.5$ mM bifurcation. It's difficult to see.

We have remade the figure and it is now much easier to see.

3.11) Fig. 6

- Please label the axes individually

See our response on Fig 5.

Appendix C

Manuscript ID RSOS-190810.R1

Title: Metabolic oscillations in *Bacillus subtilis* biofilms can be described by a minimal mathematical model

Answers to the Reviewers' comments

We thank the reviewers for their careful reading of our manuscript and their constructive and helpful comments. Our responses (blue color) to the Referee's comments are given below.

Reviewer: 2

Comments to the Author(s)

The revision was easy to follow but the authors didn't address my major concerns.

1. The authors have an asymmetry in the model between cells on the periphery and the interior of the biofilm. The reasoning of this asymmetry is based on differences in the metabolism of both groups. In the text, the authors mention that they assume that adding self-amplification on the interior wouldn't affect but they didn't demonstrate it. Adding this part in the model would allow them to not force manually the restriction of growth to cells on the periphery, being a more realistic model but maintaining the simplicity.

Answer

Thank you for the suggestion. We have now analysed an extended model including a self amplification in *Gi* and added figure S4 to demonstrate the effect of this. Nevertheless, we maintain our model in the main text to show that even a minimal model can describe biofilm oscillations and because diffusion is sufficient for the interior cells to obtain glutamate. Moreover, no dramatic difference is found in the extended model.

2. In this version of the manuscript, there are two different sets of parameters. Set A with wrong parameter values and set B with reasonable parameter values. The authors mentioned that they used set A to be able to compare their model to Wilhelm and Heinrich's model. I couldn't find this comparison. Having these two sets can confuse readers since set A contains incorrect values. Related to that, on page 8 the authors mentioned that figure 3 was created by using set A but on the figure says they used set B. I think it is necessary to eliminate set A and consider just set B, generating a new figure 3 using these values.

Answer

The new version only has a unique set of parameters, which is the one derived from experimental data. We adapted the Figures concerned accordingly.

3. Since the presented model contains only six parameters, I suggested performing a sensitivity analysis of the different parameters to see how much the results vary. The authors pointed out that in the first manuscript they already studied the two most relevant parameters. I don't think

that this is enough. An example that other parameters could also be relevant is that when they changed the constant rates $K3$ and $K4$ they needed to change $K1$ (not $k5$ or Ge).

Answer

We have made additional bifurcation plots, so now we have such plots for every parameter (see fig 5, 6 and S1 to S3). Additionally we have added Table S1 showing the unscaled and scaled sensitivities of the steady state values wrt each parameter. We have extended the paragraph on sensitivity analysis (end of p. 13) considerably, discussing Table S1.

4. I don't think that the authors' interpretation of Liu et al.'s article is correct. Liu et al. don't state biofilm oscillations would always be favorable for increasing growth rate. They were studying a particular case in which there is restriction of nutrients. The final advantage is that by using division of labor under stress, the biofilm protects the interior and the biofilm can be resistant to external attacks, such as chemicals. They didn't mention that oscillations give an advantage in the propagation of the biofilm in all conditions. In my opinion, the current text of the manuscript under revision cannot be accepted for publication.

Answer

We agree that Liu *et al.* do not state biofilm oscillations would always be favorable for increasing growth rate, and in the light of this comment, our results are more in agreement with Liu *et al.* We have now deleted the disputed sentence on p18.

5. I agree with the authors that in Liu et al. 2015 they were measuring concentration of housekeeping proteins. Nevertheless, housekeeping genes mean that these genes are constitutive, not that they have a constant concentration, especially under stress. Besides, I would like to mention that Liu et al. were considering active glutamate dehydrogenase in their model, not concentration in general (see attached original text). "The concentration of active glutamate dehydrogenase (GDH) in the interior cells (H_i); and the rate of biomass production, which is assumed to be given by the concentrations of housekeeping proteins (such as ribosomal proteins) in the interior (r_i) and the periphery (r_p)."

Answer

We changed the sentence on p3 to the following, "Ammonia and the (active form of) enzyme glutamate dehydrogenase are also variables of the model."

Giving these explanations, I hold that Garde et al. cannot make a whole point in their article criticizing the option of Liu et al. about having active GDH and concentration of housekeeping proteins as variables in the model.

Answer

We rephrased the disputed sentence on p3 as, 'Besides the quest for minimality, a reason for not considering the concentrations of proteins as variables is that they change on a longer time-scale than metabolites'. We also deleted a paragraph on p18 about this issue. In this way we do not criticize Liu's approach.

6. I agree with reviewer 3 that it would be necessary to use different colors for the different initial values in figure 4, to see the progression of the change. I don't think that it is a valid answer to say that 'Unfortunately this cannot be done in COPASI'.

Answer

We improved the said plot b using different colors.

7. Taking as an example the previous point, I found that in some cases the answers given by the authors show lack of professionalism. Also, the authors scale up the significance of their findings. I don't think this was a correct choice, since the model didn't provide new biological insights about the system.

Answer

We have made the changes more rigorously this time. We are now more careful in stressing the significance of our findings.

Minor points:

- Friedrich Schiller University Jena, Bioinformatics
- Ákos T. Kovács proof read the manuscript

Answer

Done

Reviewer: 3

Comments to the Author(s)

The authors have changed and improved the manuscript, with the remarks of Reviewers 1 and 2 often being useful. I was disappointed that many of my remarks were ignored. I suspected that other reviewers would address details of the model and biological interpretations, so I decided to point out opportunities to apply, extend, or compare results of the simple model. However, at nearly every point my remarks were denied or ignored.

Answer

We have now incorporated some more changes, see response to Reviewer 2

Indeed, perhaps some of them were beyond the scope of the paper, but why use a simple model that has already been published if you don't want to use it to contend with what's been published? For example, I pointed out that in Martinez-Corral, et al., J. Phil. Trans. R. Soc, 374, 1774 (2019) the original authors introduce their own simplification of the model. In this 2019 paper, they go through a pretty detailed comparison. Surely the authors of this submission could at least compare it? I don't think this request is beyond the scope of the paper at all.

Given that the authors expanded the paper and added interesting discussion inspired by Reviewers 1 & 2, I'm in favor of publishing the paper if the authors address how their model relates to the simplified model published in 2019 by Martinez-Corral, et al.

Answer

We now discuss the paper of Martinez-Corral in more detail on p18.

A minor remark: is it unreasonable to ask that graphs be readable? At multiple points, the authors attribute the impossibility of making graphs more readable to the graphical limitations of COPASI. I don't think this is suitable justification of not making these changes when there are so, so many freely available packages for making excellent, readable graphics. My requests along these lines were things as simple as changing axis labels. Moreover, reading the paper again, I truly think the lack of individual axis labels in the bifurcation plots makes it needlessly difficult to read the figures.

Answer

We have made new plots that are better in resolution and labeled in a way easy to read.

Appendix D

Reviewer: 2

Comments to the Author(s)

I appreciate changes in the new manuscript. I have two comments:

1) Figure 4, the units that the authors used for biomass is mmol/l (mM), and they use it to talk about growth. I would suggest converting it to cells/l since I don't think that mol ($6 \cdot 10^{23}$) per liter makes sense, considering the size of cells.

We now indicate the biomass as cells/l, as suggested. Moreover, we mention how this can be converted to g/l using a typical cell volume given in Maass *et. al.* (2011) (see p8).

2) Figure 4 and the main text about this part are confusing. The authors state that oscillating growth in biofilms is not in favor of growth rate but in figure 4 it can be appreciated that constant growth (yellow monotonic curve) is slower and doesn't reach the maximum value. In previous versions of the manuscript, the advantage of non-oscillating growth was clearer.

To verify this, we run the simulation for one more timepoint (11h) and find that the constant growth is indeed faster. The figure however, is not a good representative of the periodic halting characteristics of the growth curve and hence we choose to stick to the 10h figure. We add the 11h figure in the supplement (Fig S5).

Reviewer: 3

Comments to the Author(s)

The changes to the manuscript, especially addressing Reviewer 2's comments, have improved the paper.

Am small point I had reading the manuscript this time was the remark "We have chosen the value of the conversion factor b in Eq. (4) such that the doubling time is in agreement with the experimental values[35]" on p7 l14. This is odd because it comes much before Eq. 4. Can you just remove the reference to Eq. 4 here and then reintroduce b when Eq. 4 is defined?

Done.

The remaining issue I have is still with the readability of the graphs. In Figure 4, for example, the legend containing B1, B10, etc, while addressing previous reviewer comments, is also rather confusing. The colors themselves are also not on any kind of scale even though the parameter values they represent are. It would be good to have these, for example, as different shades on a continuum with the parameter value defined in color somewhere. I understand you are using COPASI, which apparently has limited graphical capabilities, but is this truly not possible?

Indeed this was not possible in COPASI. We created an entirely new figure 4 in R. The gradient scale indicates the initial values of G_p from 1 to 10 as suggested.

We appreciate all the valuable inputs of the reviewers that helped make the manuscript better. We also thank the reviewers for their valuable time and effort spent on the reviewing process.